# Removal of promoter CpG methylation by epigenome editing reverses *HBG* silencing

Henry W. Bell [1,7], Ruopeng Feng[2,7], Manan Shah [1], Yu Yao[2], James Douglas[2], Phillip A. Doerfler [2,3], Thiyagaraj Mayuranathan[2,4], Michael F. O'Dea [1], Yichao Li [2], Yong-Dong Wang [5], Jingjing Zhang[2], Joel P. Mackay [6], Yong Cheng [2], Kate G. R. Quinlan [1], Mitchell J. Weiss [2,7] ✉ & Merlin Crossley [1,7] ✉

β-hemoglobinopathies caused by mutations in adult-expressed *HBB* can be treated by re-activating the adjacent paralogous genes *HBG1* and *HBG2 (HBG)*, which are normally silenced perinatally. Although *HBG* expression is induced by global demethylating drugs, their mechanism is poorly understood, and toxicity limits their use. We identify the DNMT1-associated maintenance methylation protein UHRF1 as a mediator of *HBG* repression through a CRISPR/Cas9 screen. Loss of UHRF1 in the adult-type erythroid cell line HUDEP2 causes global demethylation and *HBG* activation that is reversed upon localized promoter re-methylation. Conversely, targeted demethylation of the *HBG* promoters activates their genes in HUDEP2 or primary CD34⁺ cell-derived erythroblasts. Mutation of MBD2, a CpG-methylation reading component of the NuRD co-repressor complex, recapitulates the effects of promoter demethylation. Our findings demonstrate that localized CpGmethylation at the *HBG* promoters facilitates gene silencing and identify a potential therapeutic approach for β-hemoglobinopathies via epigenomic editing.

In 1979, researchers studying the chicken globin locus noted that methylation of cytosines within CpG dinucleotides of gene promoter DNA correlated with transcriptional silencing[1]. This association was confirmed in humans, where the global demethylating drug 5-azacytidine was found to reverse silencing of the *γ-globin* genes *HBG1* and *HBG2* (hereafter referred to as *HBG)* in humans and non-human primates, thereby strengthening the correlation between CpG methylation and gene repression[2–4]. Since then, numerous studies have aimed to understand the correlation between CpG methylation and *HBG* silencing, as reversing repression of *HBG* to induce fetal hemoglobin (HbF, α2γ2) production is an established therapeutic strategy for β-hemoglobinopathies caused by mutations in the adult-expressed *β-globin* gene (*HBB)*[5–8]. However, it has been difficult to

demonstrate definitively whether the association between global CpG methylation and *HBG* gene silencing in humans is causal or correlative. Indeed, a major role for CpG methylation in *HBG* silencing has recently been brought into question[9].

In a forward genetic screen, we identified Ubiquitin-like with PHD and Ring Finger Domains 1 (*UHRF1*) as a mediator of *HBG* gene silencing. The UHRF1 protein facilitates maintenance of CpG methylation during DNA synthesis by recruiting DNA methyltransferase 1 (DNMT1) to hemi-methylated CpG sites at replication forks[10,11]. Loss of UHRF1 in adult-type *HBB*-expressing erythroid cells caused global CpG demethylation and *HBG* gene activation, while local re-methylation at the *HBG* promoters via epigenomic editing restored silencing. Conversely, *HBG* was activated by targeted promoter demethylation in the adult-

¹School of Biotechnology and Biomolecular Sciences, University of New South Wales, Sydney, New South Wales, Australia. ²Department of Hematology, St. Jude Children's Research Hospital, Memphis, TN, USA. ³Versiti Blood Research Institute, Milwaukee, WI, USA. ⁴Centre for Stem Cell Research (a Unit of inStem, Bengaluru), Christian Medical College Vellore Bagayam Campus, Vellore, India. ⁵Department of Cell and Molecular Biology, St. Jude Children's Research Hospital, Memphis, TN, USA. ⁶School of Life and Environmental Sciences, University of Sydney, Darlington, NSW, Australia. ⁷These authors contributed equally: Henry W. Bell, Ruopeng Feng, Mitchell J. Weiss, Merlin Crossley. ✉e-mail: mitch.weiss@stjude.org; m.crossley@unsw.edu.au

type erythroid cell line HUDEP2 and in primary erythroblasts derived from CD34[+] hematopoietic stem and progenitor cells (HSPCs). Taken together, our results demonstrate that CpG methylation at the *HBG* promoters causes reversible gene silencing.

## Results

### Methylation maintenance factor UHRF1 is a regulator of HbF

To identify regulatory mechanisms controlling *HBG* expression, we performed a CRISPR/Cas9 screen in the adult-type erythroid cell line HUDEP2, which primarily expresses *HBB*[12]. HUDEP2 cells expressing Cas9 were transduced with a lentiviral vector library encoding 3143 single guide RNAs (sgRNAs) targeting 776 genes encoding components of the ubiquitin proteasome system, fractionated by immune-flow cytometry according to HbF expression, and analyzed for sgRNA content by next-generation sequencing (NGS)[13]. sgRNAs that were enriched in HbF[high] cells targeted genes encoding known repressors of *HBG* transcription (*ZBTB7A*, *VHL*, *BCL11A*) and the BCL11A protein binding motif in the *HBG* promoters, which is mutated in some individuals with Hereditary Persistence of Fetal Hemoglobin (HPFH) (Fig. 1a). In addition to these controls, there was enrichment of sgRNAs targeting *UHRF1*, a DNMT1 cofactor in the CpG methylation maintenance pathway[10,14]. Lentiviral transduction of two individual *UHRF1* sgRNAs into Cas9-expressing HUDEP2 cells raised *HBG* mRNA and HbF protein levels, validating the initial screening results (Fig. 1b, c). Disruption of *UHRF1* in CD34[+] HSPCs followed by in vitro differentiation raised *HBG* and HbF in erythroid progeny to nearly the same levels as disruption of the *BCL11A* erythroid enhancer according to a clinically approved strategy[15] (Fig. 1d-f and Supplementary Fig. 1a). Consistent with its fundamental role as a DNMT1 cofactor, UHRF1 depletion caused global CpG demethylation, which accompanied the induction of *HBG* and several other genes (Fig. 1g, h). However, there were no changes in mRNAs encoding known regulators of *HBG* transcription, including BCL11A, ZBTB7A, or MYB in *UHRF1*-disrupted cells (Fig. 1h).

### UHRF1 represses *HBG* by altering the local epigenetic state

Our findings suggest that UHRF1 represses *HBG* transcription by maintaining CpG methylation, either directly at the gene promoters or indirectly. The *HBG* promoters each contain six CpG sites within positions −162 to +50 relative to the Transcription Start Site (TSS) that are highly methylated in HUDEP2 cells and in primary erythroblasts derived from in vitro differentiation of CD34[+] HSPCs (Fig. 2a and Supplementary Fig. 1b). Five days after disruption of *UHRF1* via transient expression of ribonucleoprotein (RNP) consisting of Cas9 and targeting sgRNA, overall methylation at the *HBG* promoter was reduced by ~50%, with 34% of alleles becoming demethylated at all six CpG sites (Fig. 2a, b). Coinciding with demethylation, disruption of *UHRF1* caused an increase in ATAC-Seq signal, accumulation of histone marks associated with active transcription, and recruitment of transcriptional activators GATA1 and NF-Y at the *HBG* gene promoters, all consistent with increased accessibility of chromatin (Fig. 2c). In agreement with previous reports, activation of *HBG* was associated with increased occupancy of the transcriptional repressor BCL11A, as detected by CUT&RUN (see discussion)[16,17].

Disruption of *UHRF1* had opposite epigenetic effects at the adult-expressed loci *HBB* and *HBD* (δ-globin) despite demethylation at their promoters (Fig. 2c and Supplementary Fig. 1b). Thus, the epigenetic landscape around the globin genes is directly altered by disruption of *UHRF1*, most likely via its role in maintaining CpG methylation. Moreover, global demethylation across the entire β-like globin locus caused a relative increase in *HBG* expression, consistent with the effects of demethylating drugs in β-hemoglobinopathy patients[2,3].

To study further the regulation of *HBG* by CpG methylation and UHRF1, we generated an auxin-inducible degron (AID)-tagged UHRF1 in HUDEP2 cells (Fig. 2d). Treatment with the ligand 5-Ph-IAA caused rapid depletion of UHRF1 and induction of *HBG* (Fig. 2e, lanes 1 and 2).

Washout of 5-Ph-IAA effectively restored UHRF1 levels, but *HBG* remained active with low levels of promoter methylation (Fig. 2e, lane 3, and Fig. 2f). These findings indicate that CpG methylation maintenance represses *HBG* transcription in erythroblasts. To test the effect of de novo methylation, we transfected AID-*UHRF1* HUDEP2 cells with mRNA encoding catalytically dead Cas9 (dCas9) fused to DNMT3A-DNMT3L (D3AL) or catalytically inactivated DNMT3A-3L (dD3AL) and six sgRNAs spanning the *HBG* proximal promoters from approximately −200 to +100 relative to the TSS (Supplementary Fig. 1c). Following removal of 5-Ph-IAA to restore UHRF1, directed recruitment of D3AL to the *HBG* promoters in transfected cells resulted in targeted re-methylation and silencing of transcription (Fig. 2e lane 4 and 2f). Thus, UHRF1-dependent methylation maintenance at the *HBG* promoters contributes to their transcriptional silencing.

### *HBG* is derepressed by targeted promoter demethylation

We next investigated whether targeted demethylation of the *HBG* promoters could activate transcription in adult erythroid cells. We transfected HUDEP2 cells with constructs encoding dCas9 fused to the catalytic domain of TET1 (TETv4)[18] or a catalytically compromised control variant (dTETv4) with the six sgRNAs targeting the *HBG* promoters or four control sgRNAs targeting the *EPX* gene, and purified transfected cells (Fig. 3a). Treatment of *HBG* with TETv4 resulted in effective demethylation of the *HBG* promoter CpG sites relative to controls (Fig. 3b, right panel). In contrast, only minor demethylation was observed at distal promoter CpG sites located more than 400 bp upstream of the nearest localizing sgRNA (Fig. 3b, left panel). Together, these results demonstrate that recruitment of TETv4 to the *HBG* proximal promoters confers localized demethylation of target CpG sites, which resulted in a marked increase in *HBG* expression (normalized to β-like globin expression ($HBG/(HBG + HBB)$) from 2.2% to 86% (Fig. 3c). In contrast, *HBG* increased to only 13% with dTETv4, reflecting low-level demethylation achieved by the defective catalytic domain. TETv4 'no guide' and *EPX* sgRNA treatments did not alter *HBG* expression, indicating that transcriptional activation correlates with demethylation specifically at the gene promoters.

Interestingly, high levels of *HBG* mRNA expression and HbF immunostaining cells (F-cells) were sustained over time. After three months of continuous culture following transient expression of TETv4, the %*HBG* mRNA remained >75% (Fig. 3d) and the %HbF immunostaining cells (F-cells) was >68% compared to <2% in controls (Fig. 3e). These results suggest that HUDEP2 cells lack the capacity for de novo methylation at the *HBG* promoters, which allows demethylation-induced gene activation to persist.

To confirm the responsiveness of *HBG* expression to local CpG methylation, we treated HUDEP2 cells with TETv4 or mock treatments and subsequently treated them with D3AL, catalytically inactive dD3AL, or mock treatment to see if restoring CpG methylation to the *HBG* promoters would restore gene repression. Despite slight increases in %*HBG* in controls following the double treatment protocol, TETv4-treated cells subsequently treated with D3AL substantially decreased *HBG* expression to 23%, while expression following treatment with mock or dD3AL remained at 95%. (Fig. 3f). These results demonstrate that *HBG* expression directly responds to CpG methylation levels at its promoter.

### Additive effects of individual sgRNAs and methyl-CpG sites for *HBG* induction

To better understand the relationship between *HBG* promoter methylation and gene repression, we tested the effects of TETv4 coupled with each of the six targeting sgRNAs individually, compared to treatment with all guides simultaneously, or to mock controls. Treatments with TETv4 and individual sgRNAs resulted in partial demethylation of the promoter CpGs, with a preference towards demethylation of the CpG sites nearest to the guide (Fig. 4a and

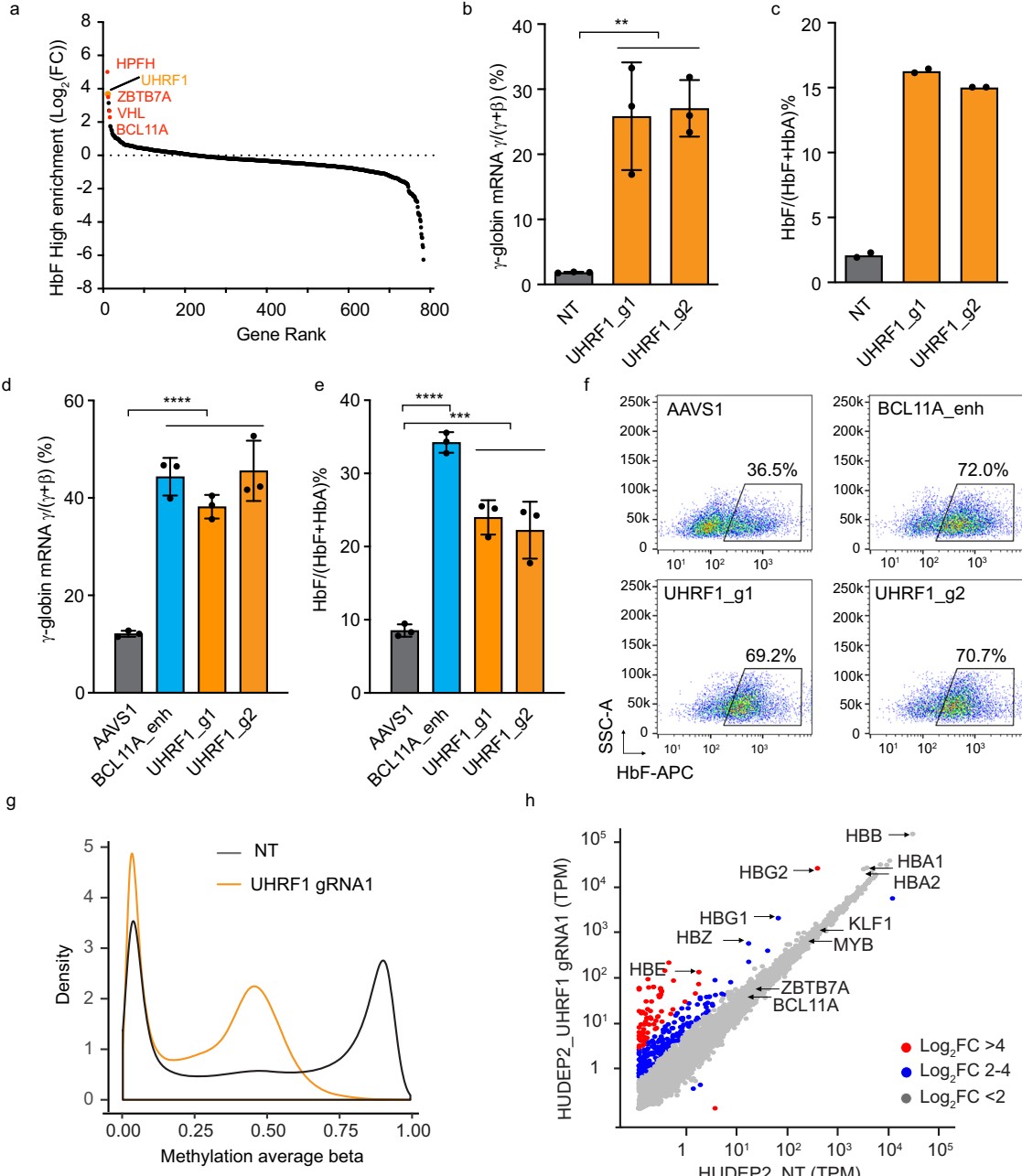

**Fig. 1 | A CRISPR/Cas9 screen identifies UHRF1 as a repressor of fetal hemoglobin (HbF) production. a** Single guide RNAs (sgRNAs) that were enriched in HUDEP2 cells expressing high levels of HbF (see Methods for details). The y-axis shows the log2 fold-change (FC) in sgRNAs comparing HbF$^{High}$ to HbF$^{Low}$ cells. Dots represent the average for four sgRNAs targeting the same gene. Positive controls for repressors of HbF, including *BCL11A*, *ZBTB7A*, *VHL*, and a BCL11A repressor binding element in the *HBG* (γ-globin) promoter (labeled HFPH) are shown. *UHRF1* is indicated as an orange point. **b** Cas9-expressing HUDEP2 cells were transduced with lentiviral vector encoding one of two different *UHRF1* sgRNAs (g1, g2) or non-targeting (NT) sgRNA control, induced to undergo erythroid maturation, and analyzed after 5 days. Graph shows *HBG* (*HBG2/HBG1*) mRNA as a fraction of total mRNA (*HBG/(HBG + HBB)*) determined by quantitative real-time qPCR (Data are presented as mean ± s.d. of three independent experiments). **c** %HbF protein (HbF/(HbF + HbA)) determined by ion exchange high-performance liquid chromatography (HPLC) analysis in two independent experiments described in panel b. Bars represent mean values. **d-f** Normal human peripheral blood CD34$^+$ cells from three

independent donors were modified by electroporation of ribonucleoprotein (RNP) consisting of Cas9 + sgRNA targeting the erythroid-specific enhancer in *BCL11A* intron 2 (BCL11A enh) or *UHRF1* protein coding regions, cultured in maintenance medium, and analyzed after 12 days. sgRNA targeting Adeno-associated virus integration site 1 (AAVS1) was used as control. **d** %*HBG* mRNA determined by qPCR. (Graph shows the mean percentage ± s.d. of three biological replicate experiments). **e** HPLC analysis of cellular hemoglobin content with %HbF indicated. (Graph shows the mean percentage ± s.d. of three biological replicate experiments) **f** Representative flow cytometry plots showing HbF immunostaining cells (F-cells). **g** Density plots showing global methylation status determined by Methylation EPIC 850 K array in *UHRF1*-disrupted or control HUDEP2 cells. **h** RNA-seq analysis showing transcripts per million mapped reads (TPM) in *UHRF1*-disrupted or control HUDEP2 cells. Each dot represents an individual gene (average of three technical replicate experiments). **P < 0.01, ***P < 0.001, ****P < 0.0001, Multiplicity-adjusted P values by one-way analysis of variance (ANOVA) for b,d,e. Source data are provided as a Source Data file.

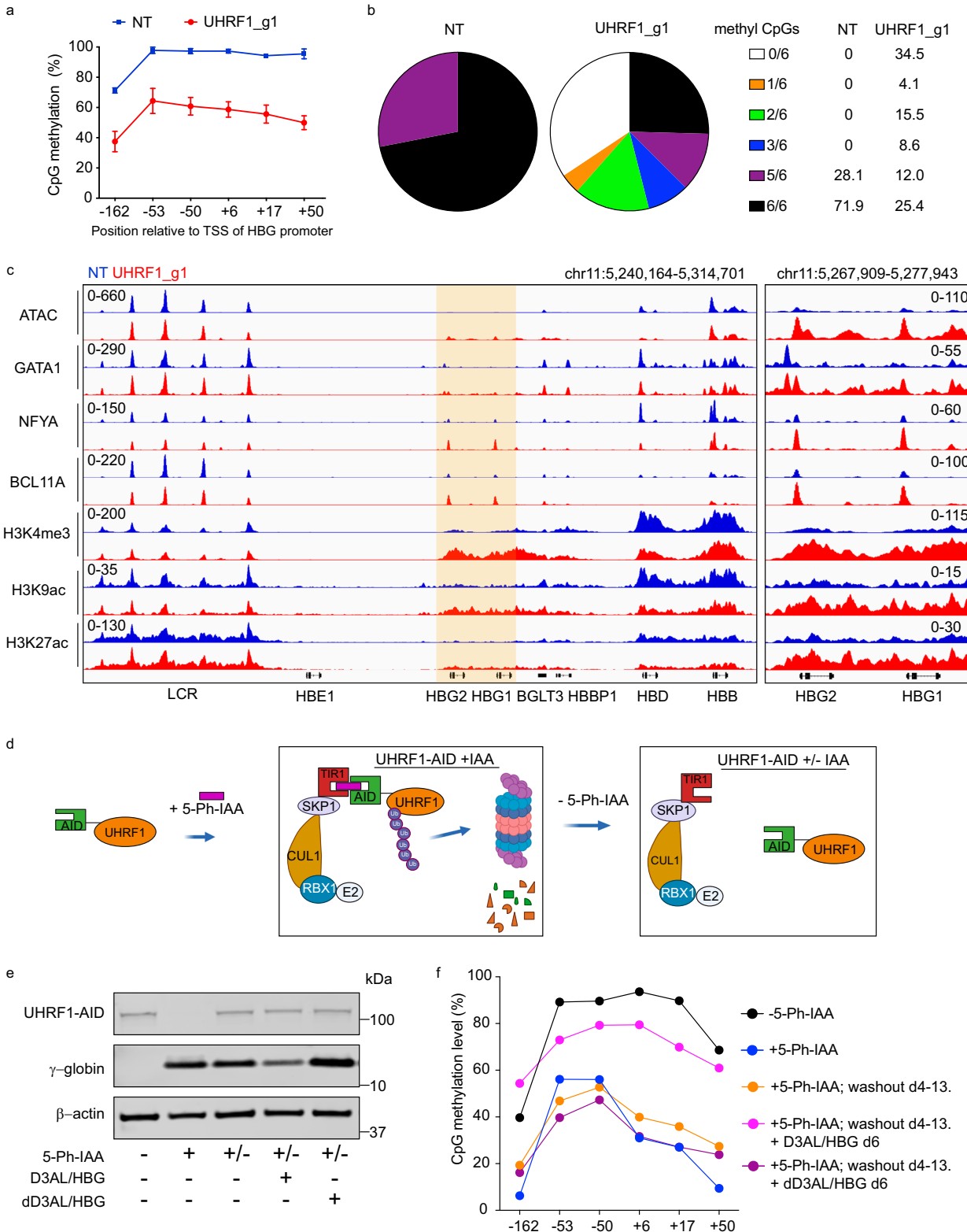

Supplementary Fig. 2a). Similarly, treatments with individual sgRNAs resulted in intermediate levels of *HBG* induction compared to simultaneous treatment with all six sgRNAs (Fig. 4b).

We leveraged the partial demethylation caused by TETv4 treatment with individual guides to investigate whether *HBG* repression is preferentially controlled by methylation of a specific subset of CpG sites. A moderate negative correlation was found between *HBG*

expression and mean methylation levels across the CpG sites surrounding the TSS from −256 to +50 (r = -0.57 (Fig. 4c, left panel). When separated based on the position of CpG sites relative to the TSS, no correlation existed between *HBG* expression and methylation of the CpG sites downstream of the TSS (r = −0.044, Fig. 4c, center panel). In contrast, a strong negative correlation with expression occurred with methylation of the four CpG sites upstream of the TSS (r = −0.72, Fig.

**Fig. 2 | CpG methylation maintenance by UHRF1 mediates *HBG* gene repression.**
**a** HUDEP2 cells were transfected with RNP consisting of Cas9 + UHRF1 sgRNA1 or NT sgRNA and grown in maintenance medium. After five days, methylation levels of individual CpGs on the *HBG* promoter were quantified by bisulfite sequencing. Graph shows the mean percentage ±s.d. of three biological replicate experiments. **b** Pie chart showing the percentage of reads containing the indicated numbers of methylated CpGs across the 6 CpG sites in the proximal *HBG* promoters or 5′ non-coding region. Each color represents a specific methylation count. **c** Genome browser screenshot showing epigenetic analysis of *UHRF1* sgRNA1-disrupted and NT sgRNA-treated CD34+ cells differentiated for 10 days. Open chromatin regions were identified by ATAC-seq. Transcription factor (GATA1, NFYA, BCL11A) occupancy and histone modifications (H3K4me3, H3K9ac, H3K27ac) were determined by CUT&RUN analysis. The *HBG1-HBG2* region highlighted in the left panel is shown

in greater detail on the right. **d** HUDEP2 cells were engineered to express UHRF1 fused to an auxin-inducible degron (AID) and OsTIR1(F74G), and the endogenous *UHRF1* gene was disrupted using Cas9. In the presence of 5-Ph-IAA, OsTIR1 combines with Skp1/Culin/F-box ubiquitin ligase components to form a functional SCF/OsTIR1 E3 ubiquitin ligase complex that degrades UHRF1-AID fusion protein. Created in BioRender. Feng, R. (2025) https://BioRender.com/9qayfxx. **e** Western blot showing the expression of UHRF1, γ-globin, and β-actin proteins in HUDEP2 UHRF1-AID cells treated as follows: Lane 2, 5-Ph-IAA × 7 days; lane 3, 5-Ph-IAA × 3 days followed by 10 day washout; lanes 4 and 5, UHRF1-AID−restored cells transduced with mRNA encoding D3AL or dD3AL + sgRNAs targeting the *HBG* promoter. **f** Methylation levels of individual CpGs near the *HBG* promoter in samples described for (**e**). Source data are provided as a Source Data file.

4c, right panel). Strong negative correlations were similarly observed at individual CpG sites within the proximal promoter but were not superior to the strength of their correlation when combined (Supplementary Fig. 2b). These results suggest that expression of *HBG* is not related to demethylation at any single CpG site, as expected if methylation affected binding of a key transcription factor to a specific motif. Rather, repression appears to occur through demethylation across multiple CpG sites in the *HBG* proximal promoters upstream of the TSS.

Next, we sought to characterize the effects of *HBG*-sgRNA-directed TETv4 treatment on the broader transcriptome of HUDEP2 cells using RNA-seq following epigenetic editing. In line with qPCR analysis, the treatment resulted in significant induction of *HBG* compared to mock treatment (log2 FC = 7.44) (Fig. 4d). Expression of other globin genes remained relatively stable, with only *HBD* (log2 FC = −0.82) and the α-globin genes (*HBA1* and *HBA2*, log2 FC = 0.64-1.85) identified within the false discovery rate (FDR) threshold of 0.01. Similar results were observed for TETv4 treatments with each guide individually (Supplementary Fig. 3a). Surprisingly, despite targeting of TETv4 to the *HBG* promoters, other genes were also identified as differentially expressed in TETv4 treatments compared to mock (Fig. 4d and Supplementary Fig. 3a).

To understand the differential expression of non-target genes, including some globin genes, we performed gene set enrichment analysis (GSEA) on a fold-change ranked gene list following TETv4 treatment. GSEA analysis identified up-regulation of erythroid cell signatures and gene ontology terms relating to chromatin organization (Supplementary Fig. 3b). Conversely, non-erythroid hematopoietic cell signatures were depleted in TETv4-treated cells. Principal component analysis did not segregate TETv4treated cells from mock, suggesting that while some gene expression changes outside of *HBG* were observed, they do not substantially alter the global transcriptome (Fig. 4e). To investigate the possibility of sgRNA-dependent off-target demethylation from TETv4, we analyzed the differential expression of genes adjacent to predicted off-target sites for each of the six targeting sgRNAs. Six differentially expressed genes were located near predicted off-target sites for guides 2 to 6 (FDR < 0.05) (Supplementary Fig. 3c, d). However, the expression of these genes was similar across all six individual guide treatments (Supplementary Fig 3c, d), suggesting that these changes are not related to guide-specific off-target effects. Together, these results suggest that the TETv4 treatments may have led to erythroid differentiation in HUDEP2 cells, resulting in differential expression of specific non-target erythroid genes through an indirect mechanism (e.g., in response to the increased *HBG* or construct expression)[19].

### Direct demethylation of promoter CpG sites activates *HBG* in CD34+ cell-derived erythroblasts

To investigate the effects of *HBG* promoter methylation on gene expression in a more physiological system, we transfected healthy donor CD34+ HSPCs with TETv4 and six *HBG* sgRNAs or controls,

followed by in vitro erythroid differentiation. Consistent with findings in HUDEP2 cells, TETv4 targeted to the *HBG* promoters resulted in demethylation of CpG sites down to a mean of ~10%, compared to ~70% with no sgRNAs or control sgRNAs (Fig. 5a). Treatment with dTETv4 and *HBG* sgRNAs resulted in intermediate demethylation (35-45%). Induction of *HBG* mRNA was roughly proportional to the level of demethylation. Thus, TETv4 directed to the *HBG* promoters resulted in approximately 35% *HBG* compared to 7-8% with control sgRNAs targeting the *EPX* gene or no sgRNA (Fig. 5b). Treatment with the catalytically impaired dTETv4 together with *HBG* sgRNAs resulted in more modest *HBG* induction, commensurate with intermediate levels of demethylation. The effects of *HBG* promoter-directed TETv4 and various controls on HbF, measured by ion exchange HPLC, were similar to changes in %*HBG* mRNA expression (Fig. 5c). Demethylation of the *HBG* promoter did not affect the expression of cell surface erythroid maturation markers CD235a, CD49d, or Band3, as measured by immune-flow cytometry (Fig. 5d). These results show that local *HBG* promoter CpG methylation contributes to gene silencing in primary adult-type erythroblasts, without obvious changes in maturation.

### CpG methylation is required for NuRD activity, but not BCL11A binding to the *HBG* promoters

To understand how changes in promoter CpG methylation regulate the *HBG* promoters, we modified CD34+ HSPCs with TETv4 + *HBG* sgRNAs, generated erythroblasts by in vitro differentiation, and used CUT&RUN to profile histone marks and chromatin occupancy by known regulators across the β-globin locus. Like observations in *UHRF1*-depleted erythroblast cells derived from CD34+ HSPCs (see Fig. 2c), *HBG* promoter demethylation was associated with increases in the activating H3K4me3 mark and recruitment of transcription factors GATA1 and NF-Y (Fig. 5e). In contrast to the effect of UHRF1 depletion, *HBG* promoter demethylation resulted in decreased H3K4me3 levels at the *HBB* promoter. Interestingly, localization of the *HBG* repressor protein BCL11A, which functions at least partly through interactions with the NuRD co-repressor complex[16,20,21], appeared to be increased at the *HBG* promoters upon TETv4 treatment, aligning with the effects of *UHRF1* disruption (see Fig. 2c). Thus, TETv4-directed CpG demethylation of the *HBG* promoters induces an active transcriptional state without impairing the binding of the key repressor BCL11A.

Given that TETv4-directed demethylation of *HBG* does not appear to disrupt repressor binding or act through a single CpG site, we tested the hypothesis that CpG methylation may be required for activity or recruitment of the NuRD complex, which interacts with methyl-CpG islands through its MBD2 subunit[22,23]. MBD2-NuRD contributes to *HBG* repression in HUDEP2 cells and is reported to directly bind the *HBG* promoter, despite its relatively low CpG density[20,24–26]. To extend these studies, we introduced a Y178F mutation into the methyl-CpG binding domain (MBD) of MBD2 to resemble more closely the paralog MBD3, which exhibits low affinity to CpG methylation and has not been associated with the regulation of *HBG* (Fig. 6a)[24,27]. By precisely modifying MBD2 at this position, we hypothesized that we could introduce

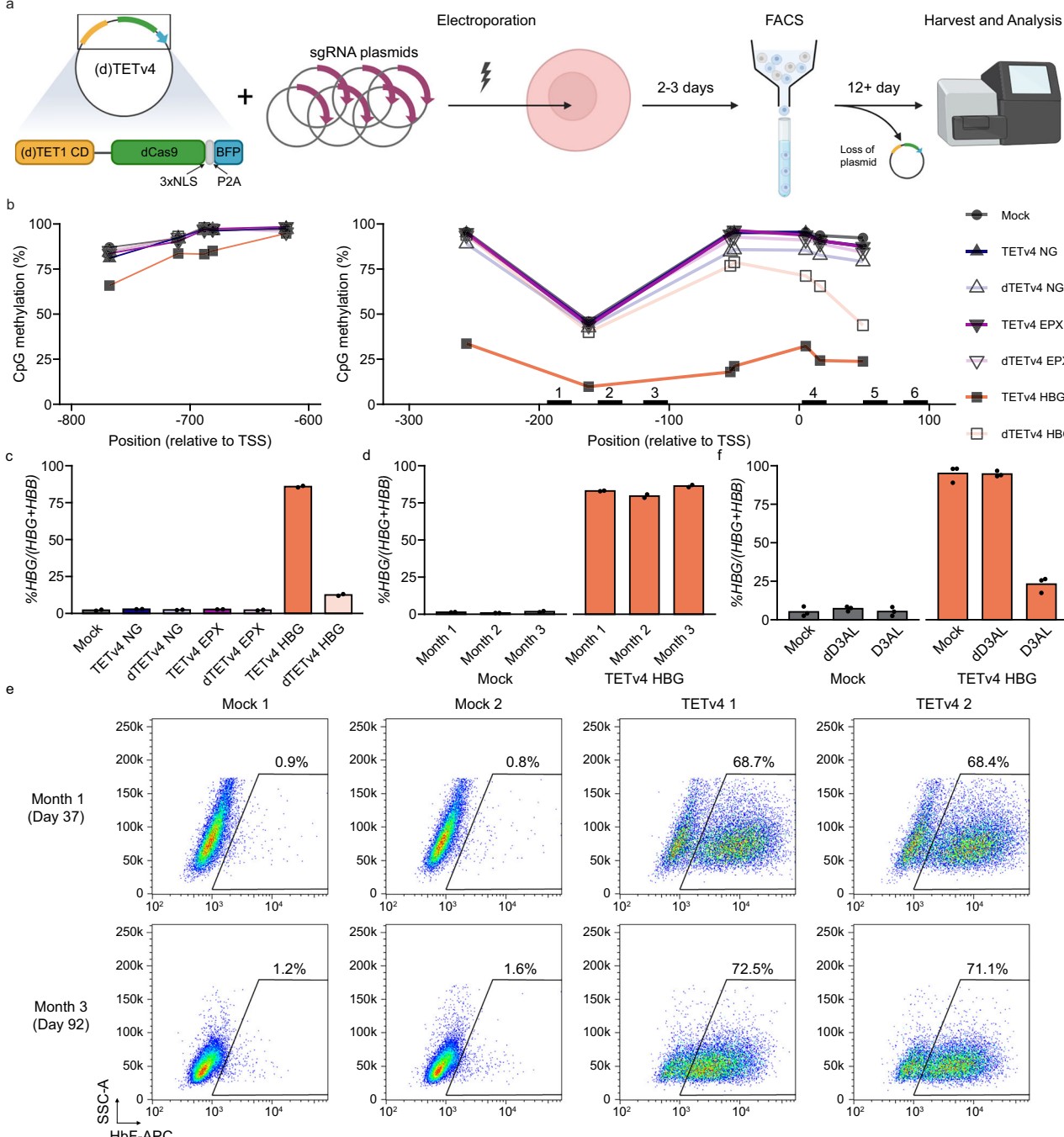

**Fig. 3 | Treatment of HUDEP2 cells with TETv4 causes localized promoter demethylation and *HBG* activation. a** Workflow for TETv4 treatments in the HUDEP2 cell line. The TETv4 construct contains a TET1 catalytic domain (CD) to remove CpG methylation linked to catalytically dead Cas9 (dCas9) and three nuclear localization signals (NLS), with blue fluorescent protein (BFP) attached by a P2A linker. dTETv4 used as a negative control, contains a mutation to depress catalytic activity. Editor and guide plasmids were introduced into HUDEP2 by electroporation and enriched by fluorescence-activated cell sorting (FACS). Cells were cultured for at least 12 days to eliminate plasmid expression before subsequent analysis. Created in BioRender. Bell, H. (2025) https://BioRender.com/ztcmiqi. **b** CpG methylation in the *HBG* distal promoter (left), and *HBG* proximal promoter and 5′ UTR (right) in TETv4-treated and mock transfected HUDEP2 cells, determined by amplicon bisulfite sequencing. Points indicate mean CpG methylation of two technical replicates, except for TETv4 EPX, where only one replicate was

analyzed. Numbered black bars indicate the positions of sgRNAs used to target *HBG*. NG = no guide. **c** *HBG* mRNA expression in HUDEP2 cells treated with TETv4 or controls, normalized to total *HBG* and *HBB* expression (*HBG/(HBG + HBB*)) measured by qPCR. Points represent measurements from two technical replicates. Bars represent mean values. **d** Serial measurements of *HBG* mRNA expression in TETv4 *HBG*-treated and control HUDEP2 lines from (**c**). at 35 days, 65 days, and 92 days post-nucleofection. Two technical replicates were analyzed across three time points. **e** Flow cytometry plots showing HbF staining (F-cells) in mock and TETv4 *HBG*-treated HUDEP2 lines at day 37 (top) and day 92 (bottom). **f** Relative *HBG* mRNA expression in HUDEP2 cells treated with TETv4 following subsequent treatment with D3AL and controls. Expression normalized to total *HBG* and *HBB* expression (*HBG/(HBG + HBB*)) measured by qPCR. Points represent measurements from three technical replicates. Bars represent mean values. Source data are provided as a Source Data file.

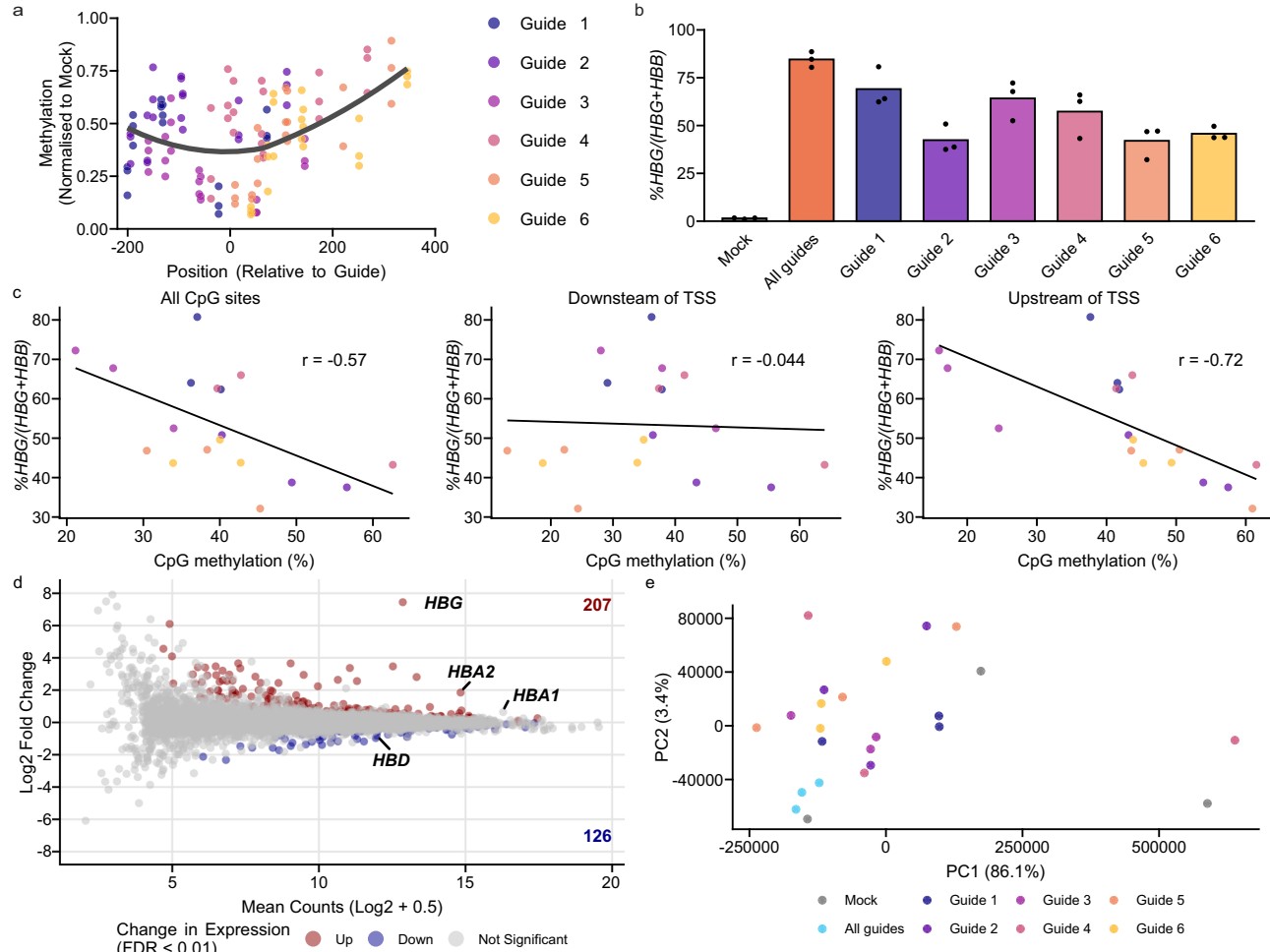

**Fig. 4 | *HBG* expression correlates with CpG demethylation in the proximal promoter. a** Relationship between the distance of the TETv4 target site to CpG sites within the *HBG* proximal promoters and 5′ UTR, and demethylation activity at those sites following treatment with TETv4 and *HBG* guides 1-6 individually. Methylation (Y-axis) represents the methylation % in each sample divided by the mean of the methylation % in the mock-treated samples at each site. Position (X-axis) indicates the distance between the CpG site and the center of the sgRNA sequence. Trend line was generated using the LOESS method with span = 1. Points indicate three technical replicates measured at each site and colored by the sgRNA used for TETv4 treatment. **b** *HBG* mRNA expression in HUDEP2 cells treated with TETv4 + *HBG* sgRNAs 1-6 individually or all together. Points represent measurements from three technical replicates. Bars represent mean values. **c** Pearson's correlation between the mean methylation % from a. at all CpG sites across the *HBG*

proximal promoter and 5′ UTR (positions −256, −162, −53, −50, +6, +17, +50; left), sites downstream of the transcription start site (TSS) only (positions +6, +17, +50; center) and sites upstream of the TSS only (positions −256, −162, −53, −50; right), vs. *%HBG* mRNA from b. in HUDEP2 cells treated with TETv4 and *HBG* sgRNAs 1–6 individually. Trendline fit was determined using the least squares method. **d** MA plot of differentially expressed genes in TETv4 *HBG*-treated *vs.* mock-treated HUDEP2 cells (3 replicates per group). Colored points represent genes identified as upregulated (red, 207 genes) or downregulated (blue, 126 genes) with a false discovery rate (FDR) cutoff of 1% (Benjamini-Hochberg method). *HBG* represents transcripts mapping to both *HBG1* and *HBG2*. **e** Principal component (PC) analysis of transcript expression in HUDEP2 lines treated with mock or TETv4 + *HBG* sgRNAs 1–6 individually or all together. Source data are provided as a Source Data file.

a methyl-CpG binding deficiency into the protein, without affecting its other functions.

To confirm that the proposed Y178F mutation inhibited MBD2's ability to bind methylated CpG sites at the *HBG* promoters, we generated MBD2-GATAD2A coiled-coil recombinant proteins with or without the Y178F mutation (MBD2sc Y178F/WT) and tested their ability to bind the *HBG* −53/−50 CpG sites or a positive control based on the CpG-rich BCAT1 promoter[28] in vitro by electrophoretic mobility shift assay (EMSA) (Fig. 6b, c). Like the positive control, MBD2sc WT only bound the methylated *HBG* −53/50 CpG sites, confirming the importance of CpG methylation for MBD2 binding (Fig. 6c). In contrast, MBD2sc Y178F was unable to form a stable complex with CpG methylated probes, despite comparable loading and stability of the expressed proteins (Fig. 6c, Supplementary Fig. 4a). These results demonstrate that the MBD2 Y178F mutation markedly affects binding to methylated CpG sites, including at the *HBG* promoters.

To investigate the effect of MBD2 Y178F on *HBG* regulation, we introduced the mutation into the endogenous *MBD2* gene using CRISPR/Cas9 homology-directed repair (HDR) and selected five HUDEP2 clones expressing MBD2 Y178F at similar levels to WT clones for analysis (Supplementary Fig. 4b). RNA-seq analysis comparing MBD2 Y178F lines to WT revealed a clear induction of *HBG*, with log2 fold change of 2.7. Only 33 other genes were identified as differentially expressed with in MBD2 Y178F mutant lines (FDR < 0.01), including hematopoietic regulators *GATA2 and HHEX*, which are weakly linked with *HBG* activation, and *RUNX1*, which may be a repressor[29–32] (Fig. 6d). In each case, differential expression associated with the Y178F mutation was in a direction that would be expected to dampen *HBG* expression, with log2 fold changes of −0.75 and −0.68, and 0.84 for *GATA2*, *HHEX*, and *RUNX1*, respectively. Thus, differential expression of these genes is unlikely to contribute to the induction of *HBG* expression (Fig. 6d). Principal component analysis of transcript

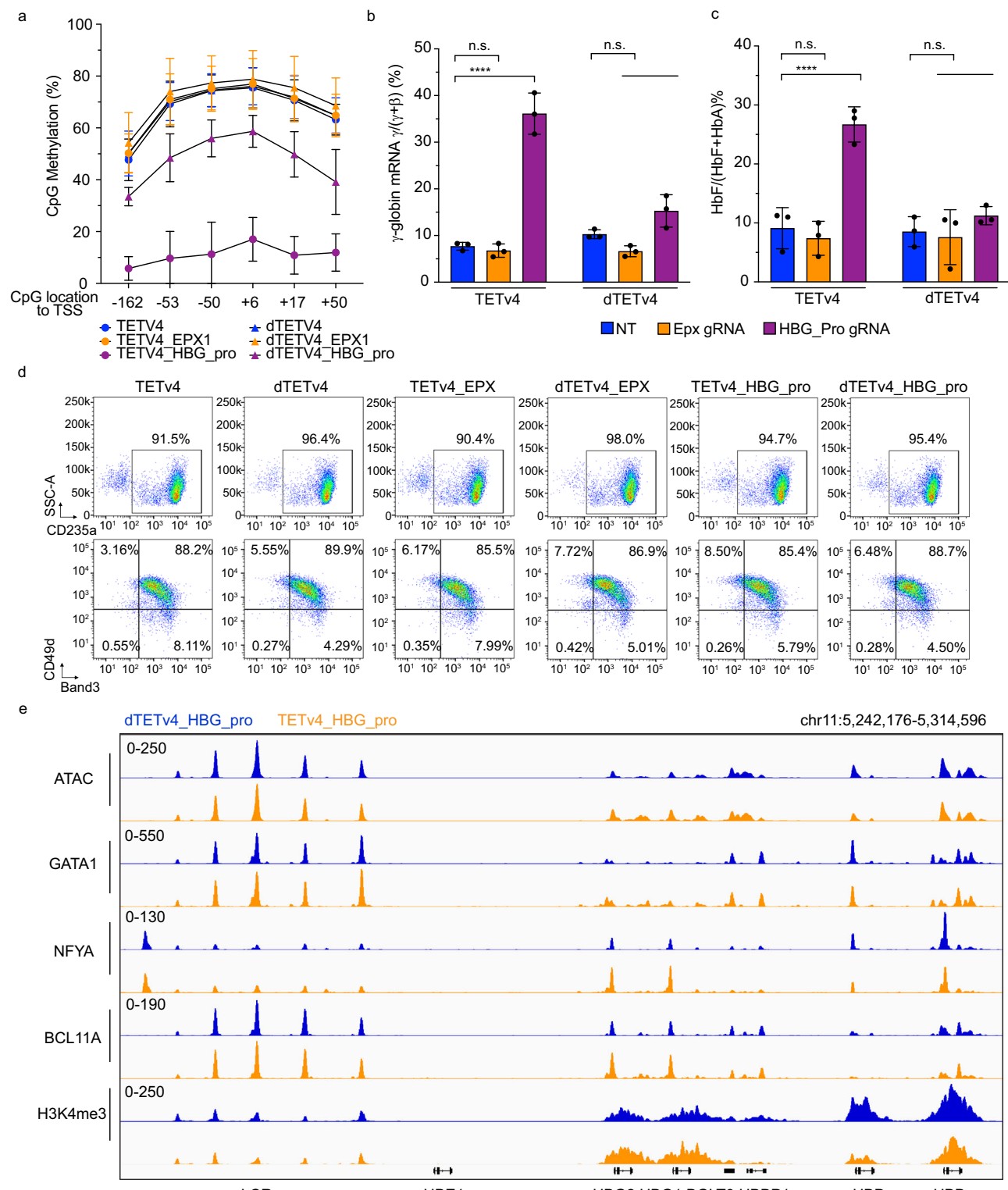

**Fig. 5 | *HBG* promoter demethylation activates gene expression in erythroblasts derived from CD34⁺ cells.** Healthy donor CD34⁺ cells were electroporated with TETv4 mRNA or dTETv4 mRNA + six sgRNAs targeting the *HBG* promoter or sgRNA targeting a control gene (*EPX*), followed by in vitro erythroid differentiation. **a** Bisulfite Amplicon sequencing at day 12 of differentiation showing methylation levels at the indicated CpG sites flanking the *HBG* promoter. Data show mean percentage ± s.d. of three biological replicates. **b** %*HBG* mRNA following induced erythroid differentiation. Data show mean percentage ±s.d. of three biological replicates. **c** %HbF determined by ion exchange HPLC following induced differentiation. Data show mean percentage ± s.d. of three biological replicates. **d** Representative FACS profiles of erythroid maturation markers CD235a, CD49d, and Band3 at day 15 of differentiation. **e** Epigenetic analysis of the extended β-like globin locus in TETv4-treated or control dTETv4-treated erythroblasts on day 10 of erythroid differentiation. ATAC-seq analysis indicates chromatin accessibility. CUT&RUN analysis shows the occupancy of GATA1, NFYA, and BCL11A, and H3K4me3 modifications. ****$P < 0.0001$, Multiplicity-adjusted P values by one-way analysis of variance (ANOVA) for (**b**, **c**). Source data are provided as a Source Data file.

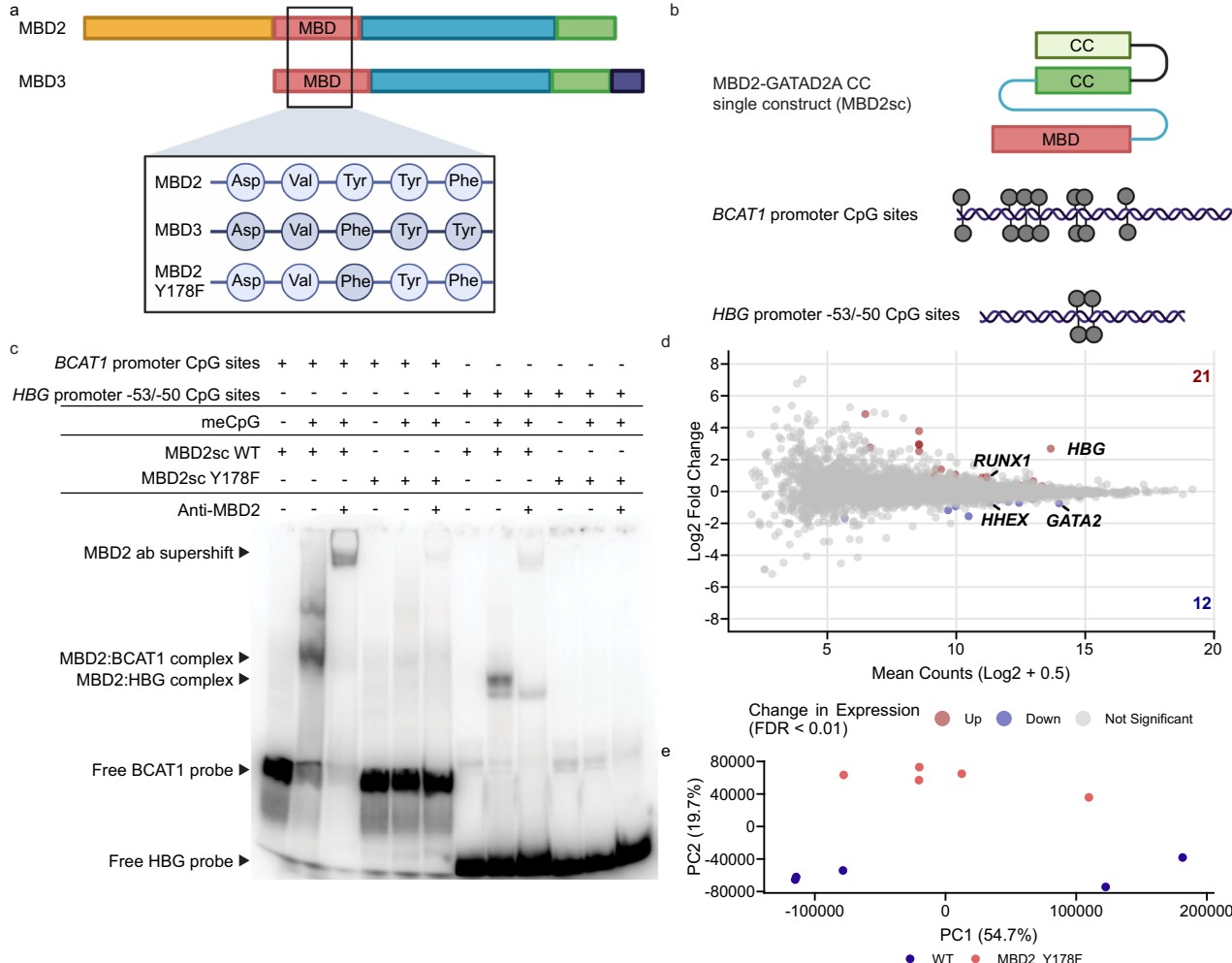

**Fig. 6 | Mutation of the MBD2 methyl-CpG binding domain impairs *HBG* repression. a** Schematic of the MBD2 and MBD3 protein sequences showing the methyl-CpG binding domains (MBD) and the location of the MBD2 Y178F mutation to create an MBD3-like domain with impaired methyl-CpG binding. Created in BioRender. Bell, H. (2025) https://BioRender.com/ztcmiqi. **b** Diagram of components used in EMSA analysis. The MBD2 single construct (MBD2sc) contains the MBD (red), intrinsically disordered region (IDR, blue), and coiled-coil (CC, dark green) of MBD2 linked to the CC of GATAD2A (light green). Target probes for the *BCAT1* and *HBG* promoters are shown with CpG sites (gray pins) indicated. Created in BioRender. Bell, H. (2025) https://BioRender.com/ztcmiqi. **c** Electrophoretic mobility Shift assay (EMSA) of probes representing *BCAT1* or *HBG* promoter CpG sites complexed with WT and Y178F MBD2sc expressed protein from b. + or - signs represent the presence or absence of each component in the lane. Arrows indicate the positions of free probe, MBD2:probe complexes, and antibody:MBD2:probe (ab) complex supershifts. **d** MA plot of differentially expressed genes in MBD2 Y178F mutant *vs.* WT HUDEP2 cells (five clones per group). Colored points represent genes identified as upregulated (red, 21 genes) or downregulated (blue, 12 genes) with a false discovery rate (FDR) cutoff of 1% (Benjamini-Hochberg method). *HBG* represents transcripts mapping to both *HBG1* and *HBG2*. **e** Principal component (PC) analysis of transcript expression in MBD2 Y178F mutant and WT HUDEP2 lines. Source data are provided as a Source Data file.

expression identified MBD2 Y178F mutant lines clustered with WT lines on the first principal component, while only separating on the second principal component, suggesting that transcriptional differences induced by the MBD2 Y178F mutation are relatively smaller than the variation within the HUDEP2 cell line used (Fig. 6e). Together these results align with recent studies[25,26] and suggest that the *HBG* genes are a uniquely important target of MBD2-NuRD, which requires CpG methylation to repress transcription in adult-type erythroid cells.

## Discussion

Inducing HbF production during adult erythropoiesis is an effective strategy for treating β-hemoglobinopathies[33]. Preclinical and clinical studies over almost 50 years have established a strong correlation between demethylation of the *HBG* promoters and their transcriptional activation, although the associated mechanism has remained elusive. Our studies advance the field by demonstrating that methylation of the *HBG* promoters directly silences their expression in

erythroid cells, and demethylation alleviates silencing. These discoveries were facilitated by the recent availability of sgRNA-directed Cas9-mediated epigenome editing tools that can install or remove epigenetic marks at regions of interest[18].

Interestingly, while genome-wide hypomethylation via *UHRF1* disruption or mutational inactivation of the global CpG-methylation reader MBD2 in erythroblasts caused increased *HBG* transcription, relatively few other genes were significantly derepressed by these perturbations, congruent with the relatively minor phenotype of *MBD2* knock-out mouse models[34,35]. Moreover, although loss of *UHRF1* caused demethylation over the entire β-like globin locus, including the adult *HBB* gene, *HBG* was most strongly induced. These findings indicate that the fetal *HBG* genes are important targets for CpG methylation-induced silencing during erythropoiesis and are especially poised for activation by demethylation.

Establishing that methylation of the *HBG* promoters is required for developmental gene silencing provides further opportunities to

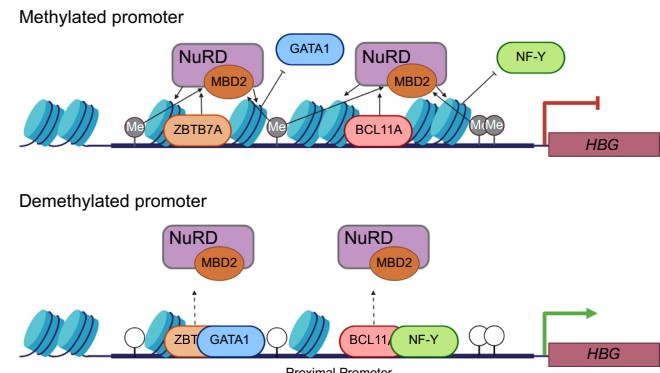

**Fig. 7 | Model for repression of the *HBG* genes by methylation of promoter CpG sites.** To repress transcription, the MBD2-NuRD complex interacts with BCL11A, ZBTB7A, and methylated CpG sites (Me) within the proximal *HBG* promoter and directs the remodeling of nucleosomes to exclude the binding of activating transcription factors GATA1 and NF-Y. In the absence of CpG methylation, interaction of the NuRD complex with repressive transcription factors alone is not sufficient to establish the repressive positioning of nucleosomes within the proximal promoter, facilitating the preferential binding of activating transcription factors to drive gene expression. Created in BioRender. Bell, H. (2025) https://BioRender.com/ztcmiqi.

examine the associated mechanisms. Collectively, previous findings and the current study support a model in which *HBG* gene silencing by promoter CpG methylation is one of several mechanisms that act additively or synergistically to recruit NuRD[21,25,26,36,37]. Various perturbations can derepress *HBG* during adult erythropoiesis, including reducing BCL11A or ZBTB7A expression, inhibiting the binding of those repressors to the *HBG* promoters, demethylation of key promoter CpGs, altering NuRD subunits, including the methyl-CpG reader MBD2 or the catalytic subunit CHD4, and *DNMT1* mutation or inhibition[25,38–46].

Interestingly, activation of *HBG* by UHRF1 depletion or TETv4-directed promoter demethylation was associated with increased occupancy by the transcriptional repressor BCL11A. The apparent inconsistency of increased repressor binding at an active gene may be explained by improved access of BCL11A (or its CUT&RUN antibody) to the open chromatin of the transcriptionally active promoter[16,17]. The persistence of gene expression in this instance indicates that binding of BCL11A to the promoter is insufficient to fully repress transcription. Observations that BCL11A occupancy persists at the *HBG* promoters despite reactivation of the genes by CpG demethylation or ZBTB7A disruption[16], and that BCL11A depletion combined with ZBTB7A depletion or CpG demethylation induces transcription additively[9,36], suggest that no single mechanism is dominant and that numerous approaches alone or in combination can be used to induce HbF therapeutically.

Our results support associations between the density and location of CpG methylation for gene repression, with the proximal promoter upstream of the TSS being the most important region. Although our analysis found strong associations between methylation of the *HBG* proximal promoter and gene repression, no individual site displayed primacy compared to the collective methylation across this region. The view that no single methyl-CpG site is responsible for silencing fits with studies on naturally occurring HPFH-associated *HBG* promoter mutations, which have never implicated a lone CpG in gene silencing, despite the fact that CpGs are mutational hotspots[47]. In contrast, human genetic studies have uncovered multiple HPFH mutations in the BCL11A or ZBTB7A binding motifs[38]. Moreover, we find no evidence for overlap between *HBG* promoter CpG sites and cognate motifs for key transcription factors. Thus, CpG methylation does not appear to repress *HBG* by altering transcription factor binding directly, which

represents one proposed mechanism for epigenetic regulation of gene expression[48]. In contrast, the requirement of multiple methylated CpG sites for effective *HBG* repression is consistent with the function of the MBD2-NuRD complex, which is thought to mediate transcriptional repression by associating with high-density regions of CpG methylation, including CpG islands[23]. High methyl-CpG density has been found to trap MBD2 within its target regions as it moves rapidly between CpG sites, which may guide the placement of repressive nucleosome structures within the region of persistent occupancy[49]. Although the CpG density in the *HBG* promoters is below that expected to effectively trap MBD2 on its own, it has been proposed that transcription factor-mediated recruitment may be able to supplement CpG density and enable sufficient MBD2-NuRD occupancy for gene silencing[49]. In support of this theory, CpG sites associated with *HBG* silencing are positioned near independent binding motifs for the transcriptional repressors BCL11A and ZBTB7A, which both interact with NuRD[21,25,26,36,38,39]. Moreover, disruption of the interaction between BCL11A and NuRD results in impaired chromatin remodeling and activation of the *HBG* genes, highlighting the importance of this interaction for NuRD function[16]. Our results extend this concept by highlighting the importance of CpG methylation at the *HBG* promoter for repression, while further demonstrating, in agreement with recent findings, the importance of the methyl-CpG-MBD2-NuRD interaction for *HBG* repression ([26] and this study).

Collective evidence supports a model in which CpG methylation operates in concert with BCL11A and ZBTB7A occupancy to stably recruit MBD2-NuRD to the *HBG* promoters, which remodels local nucleosome structure to establish silencing, in part by inhibiting the recruitment of transcriptional activators GATA1 and NF-Y (Fig. 7). In the absence of CpG methylation to stabilize MBD2-NuRD occupancy, relaxed nucleosome arrangements may permit binding of transcriptional activators to drive *HBG* expression without inhibiting repressor binding, which may also benefit from increased accessibility. Our work confirms the importance of CpG methylation in the *HBG* promoter region for transcriptional repression, while furthering the understanding of how CpG methylation in this region achieves silencing. Additional studies are required to investigate the dependency of nucleosome positions in the *HBG* promoters on methylated CpG and transcription factor binding sites to confirm the precise details of this mechanism.

Finally, our findings support the longstanding claim that inhibition of CpG methylation can be adapted therapeutically for β-hemoglobinopathies, which is further supported by the recent work of Amistadi et al.[50]. Drugs that cause global demethylation of CpG sites, including 5-azacytidine and DNMT1 inhibitors, can induce HbF to therapeutic levels[3,46,51]. However, these approaches are non-specific and have toxicities that are likely to interfere with long-term use. In principle, targeted demethylation of the *HBG* promoters by epigenetic editing could circumvent these problems. While TETv4-mediated demethylation of the *HBG* promoters and consequent transcriptional activation were sustained in HUDEP2 cells for many weeks in culture, the duration of this effect in vivo remains to be established. It is uncertain whether targeted demethylation of the *HBG* promoters in hematopoietic stem cells by one-time treatment with TET1-based editors could be sustained for a lifetime. However, with sufficient efficiency and persistence of demethylation, it may be feasible to induce HbF therapeutically via periodic, targeted delivery of an epigenomic editor to the *HBG* genes in erythroid precursors. The use of epigenetic editors has theoretical safety advantages over conventional CRISPR/Cas9 editing and base editing, which both rely on error-prone DNA repair mechanisms to resolve DNA breaks and mismatches[52,53]. In contrast, epigenome editing utilizes catalytically inactive Cas9 that cannot cleave DNA, thereby leaving the genome intact and avoiding potentially harmful gene deletions and rearrangements.

## Methods

### HUDEP2 cell culture

Human umbilical cord blood derived 2 (HUDEP2) cells were grown in serum free expansion medium (SFEM, StemSpan #9650) supplemented with 50 ng/mL stem cell factor (R&D Systems, #255-SC-200/CF), 3 IU/mL erythropoietin (PeproTech, #100-64-100), 10 µM dexamethasone (Sigma-Aldrich, #D4902), 1 µg/mL doxycycline (Sigma-Aldrich, #D9891), and 0.5 mg/mL PSG (Thermo Fisher Scientific, #10378016). All cell cultures were incubated at 37 °C in a water-jacketed incubator with 5% $CO_2$.

Differentiation medium for HUDEP2 cells consisted of Iscove's Modified Dulbecco's Medium (IMDM) (Thermo Fisher, #12440053), 2% fetal bovine serum (FBS), 5% inactivated human plasma (SeraCare, #1810-0008), 330 µg/mL human holo-transferrin (Sigma-Aldrich, #T0665), 10 µg/mL human insulin (Sigma-Aldrich, #I9278), 2 IU/mL heparin (Sigma-Aldrich, #H3149), 3 IU/mL erythropoietin (Amgen, #55513-144-10), 50 ng/mL SCF, 1 µg/mL doxycycline, and 1% penicillin–streptomycin (Gibco, #15070-068) solution.

### CRISPR/Cas9 screen for genes that regulate HbF production

A pooled lentiviral sgRNA library targeting 776 genes encoding ubiquitin ligases, accessory proteins, and deubiquitinases was prepared in the lentiviral vector lentiGuide-Puro (Addgene plasmid 52963)[54]. Approximately $3.5 \times 10^6$ HUDEP2 cells stably expressing Cas9 were transduced with the lentiviral sgRNA library at a multiplicity of infection of 0.3. After 24 h, transduced cells were cultured in puromycin (1 µg/mL) (Gibco, #A1113803) for 3 days and then expanded in HUDEP2 maintenance medium for 5 days. Erythroid maturation was induced by switching to differentiation medium for 5 days, followed by staining with an anti-fetal hemoglobin (HbF) monoclonal antibody conjugated to APC. Cell populations with the 10% highest and lowest anti-HbF staining intensities were purified by fluorescence-activated cell sorting (FACS). Genomic DNA from sorted cells was extracted using the DNeasy Blood and Tissue kit (Qiagen). PCR amplicons were generated to identify integrated sgRNAs in the HbF[high] and HbF[low] fractions, and products were separated on 2% agarose gels. DNA bands of the expected size were excised and analyzed by MiSeq 300-bp paired-end sequencing (Illumina, #MS-102-2002). Analysis of sgRNA distributions was performed using the Model-based Analysis of Genome-wide CRISPR/Cas9 Knockout (MAGeCK) method (version 0.5.9.4)[55].

### CD34+ HSPC culture and in vitro erythroid differentiation

Human G-CSF-mobilized mononuclear cells were purchased from StemExpress (Folsom, CA, USA). The CD34+ cells were enriched using a CliniMACS system in the Human Applications Laboratory at St. Jude Children's Research Hospital (St. Jude; Memphis, TN, USA), and cryopreserved in liquid nitrogen[56]. Cells were thawed and grown in expansion medium consisting of supplemented X-VIVO 10 media (Lonza, #BEBP02-055Q) with 100 ng/mL human Flt3-Ligand (R&D systems, 308-FKHB-010), 100 ng/mL stem cell factor (SCF), 100 ng/mL thrombopoietin (R&D systems, 288-TP/CF), 1× penicillin/streptomycin, and 2 mM GlutaMax™, (Gibco, 35050061) for 1–2 days.

Erythroid differentiation of CD34+ cells was performed using a three-phase culture protocol. For phase 1 (days 0–7), cells were maintained at a density of $1–4 \times 10^5$/mL in IMDM supplemented with 2% human AB plasma, 3% human AB serum, 200 µg/mL holo-transferrin, 3 IU/mL heparin, 1 IU erythropoietin, 10 ng/mL SCF, and 1 ng/mL interleukin-3 (IL-3)(R&D systems, 203-IL/CF), 1% penicillin–streptomycin. For phase 2 (days 8–13), IL-3 was removed from the Phase 1 medium, and the cells were maintained at a density of $2–8 \times 10^5$/mL. For phase 3 (days 13–18), IL-3 and SCF were omitted, the holo-transferrin concentration was increased to 1 mg/mL, and the cells were maintained at a density of $1–2 \times 10^6$/mL.

Erythroid maturation was monitored by flow cytometry analysis by using FITC-conjugated anti-CD235a (BD Biosciences, #559943,

clone GA-R2), APC-conjugated anti-Band3 (Gift from Dr. Xiuli An), and Violet 421 anti-CD49d (Biolegend, #304322, clone 9F10) antibodies.

### Gene disruption in CD34+ HSPCs

CD34+ HSPCs were thawed into expansion medium 24 h before electroporation. For gene disruption, ribonucleoprotein complexes (RNPs) were formed by incubating Cas9 3×-NLS (50 pmol) and sgRNA (150 pmol) for 20 min at room temperature then, mixed with $2 \times 10^5$ cells in 20 mL of room temperature P3 buffer (Lonza, # V4SP-3096) and electroporated with the Lonza 4D-Nucleofector System (electroporation program DS-130). After electroporation, cells were grown in CD34+ cell erythroid differentiation medium phase I. Editing frequencies were determined by targeted amplicon sequencing followed by analysis with CRIS.py (version 1)[57].

### Hemoglobin quantification

Cells were lysed in water, centrifuged, and hemoglobins in the supernatant were quantified by high-performance liquid chromatography (HPLC), using a PolyCAT A ion-exchange column (PolyLC Inc., #202CT0510) on a Prominence HPLC System (Shimadzu Corporation) with LabSolutions Software, v.5.81 SP1. The eluted proteins were identified by light absorbance at 418 nm using a diode array detector. The relative amounts of different hemoglobins were calculated from the areas under the peaks and normalized to a wild-type control.

### CRISPR/Cas9 gene disruption of UHRF1 in HUDEP2 cells

HUDEP2 cells stably expressing Cas9 and maintained in expansion medium were transduced with lentiGuide-Puro vector (Addgene, plasmid #52963) targeting sgRNA for UHRF1 (Supplementary Table 4) or control nontargeting sgRNA. After 24 h, 1 mg/mL puromycin (Thermo Fisher Scientific, #A1113803) was added. Transduced cells were enriched by selection in puromycin (1 µg/mL) for 3 days, and erythroid maturation was induced by switching the cells to HUDEP2 differentiation medium for another 5 days.

### Generation of the UHRF1-Auxin-induced degron (AID) HUDEP2 cell line

We used the AID2 system[58] to generate HUDEP2 cells with auxin-degradable UHRF1 (UHRF1-AID HUDEP2 cells). Wild-type HUDEP2 cells were transduced with lentiviral vector pBlast encoding UHRF1 fused to an AID at the carboxyl terminus, then the endogenous *UHRF1* gene was disrupted by transfection of RNP containing Cas9 + targeting sgRNAs. The bulk population of transfected cells was transduced with pBlast OsTIR1(F74G) lentiviral vector. Clones were isolated by FACS sorting and expanded in maintenance medium. Auxin-induced degradation of UHRF1-AID was induced by the addition of 1 µM 5-Ph-IAA (MedChemExpress, #HY-134653) and verified by western blotting with an anti-UHRF antibody after 1–3 days.

### Epigenome editing in HUDEP2 cells

The TETv4 plasmid, containing the Ten-Eleven Translocase methylcytosine dioxygenase 1 (TET1) catalytic domain (TET1CD) fused to a XTEN80 linker, catalytically dead Cas9 (dCas9) and P2A-linked blue fluorescent protein (BFP) (Addgene plasmid # 167983), and CRISPRoff-v2.1, containing the active domains of DNMT3A and DNMT3L fused to an XTEN80 linker, dCas9, BFP, and KRAB (Addgene plasmid # 167981) were kindly provided by Luke Gilbert[18].

A variant of TETv4 was created by overlap extension PCR and restriction enzyme (RE) cloning to introduce mutations at +760C>T and +767/768 AC>CT, which depress catalytic activity of the TET1CD[59,60] (dTETv4).

CRISPRoff-v2.1 was modified to remove the KRAB domain, with the resulting construct named D3AL in reference to the key functional domains. A variant of D3AL was created by overlap extension PCR and RE cloning to introduce a +299G>C mutation to disrupt the catalytic

activity of the DNMT3A catalytic domain[61] (dD3AL). An additional variant was created, replacing BFP with EGFP. Modified epigenetic editing constructs are available upon request.

sgRNA sequences were designed using a CRISPR design tool (Benchling) and selected based on on-target score and their spacing around the TSS. Guides used in epigenetic editing are listed in Supplementary Table 1.

For TETv4-directed epigenetic editing in the HUDEP2 cell line, sgRNAs were expressed from transfected plasmids. Briefly, a pcDNA3 plasmid containing the mCherry fluorophore (a gift from Beeke Wienert) was modified to contain the sgRNA expression scaffold from PX458. Guide sequences were synthesized as single-stranded oligonucleotides (IDT) containing sticky ends compatible with the BbsI sites of the PX458 sgRNA scaffold and inserted into pcDNA3-mCherry-sgRNA plasmid digested with BbsI (New England Biolabs (NEB), #R3539S). pcDNA3-mCherry-sgRNA plasmids for each target gene were combined at equal molar ratios and to a final concentration of 500 ng/mL. HUDEP2 cells were washed in 1× PBS before transfection using the Neon electroporation system (Invitrogen) with 600 ng TETv4 or dTETv4, and 500 ng of premixed pcDNA3-mCherry-sgRNAs per $3 \times 10^5$ cells in 1× buffer T. Each transfection was performed three times using single pulses of 1100, 1200, and 1300 V for 20, 30, and 40 ms respectively. For no guide (NG) and mock controls, transfections were performed in the absence of pcDNA3-mCherry-sgRNA with and without the editor plasmid, respectively. Each treatment was repeated on two or three samples from the parent population. Following 48–72 h of recovery from transfection, FACS was performed using the FACSMelody (BD Biosciences) to sort live cells (expressing the endogenous Kusabira Orange fluorescent mark (KO)) that had taken up the TETv4 and pcDNA3-mCherry-sgRNA plasmids (KO+/BFP+/mCherry+) into a pool, selecting for successfully transfected cells. The pooled populations were maintained for at least 14 days post-nucleofection to deplete the transfected plasmids before downstream analysis.

For D3AL-directed epigenetic editing in the HUDEP2 cell line, D3AL and dD3AL editors were transfected as mRNA along with chemically synthesized sgRNAs (IDT or Synthego). Transcripts for mRNA synthesis were prepared by PCR with primers to attach the T7 promoter (Supplementary Table 2). In vitro transcription (IVT) was performed using the HiScribe® T7 mRNA Kit with CleanCap® Reagent AG (NEB, #E2080S) fully substituted with N1-Methylpseudo-UTP (Thermo Fisher Scientific, #R0491SKB007) and supplemented with 1 U/mL SuperRNasin (Thermo Fisher Scientific, #AM2694). PolyA tailing was performed using 25 U of *E. coli* Poly(A) Polymerase (NEB, #M0276S) before precipitating mRNA by the addition of 20 μl of the included LiCl solution. HUDEP2 cells were washed in 1× PBS before transfection using the Lonza 4D nucleofection system (electroporation program DS−150) with 1 μg D3AL or dD3AL, and 2 μg of premixed sgRNAs per $5 \times 10^5$ cells in 1× buffer P3. Mock controls were electroporated in the absence of substrate. Successfully transfected GFP+ cells were purified by FACS and expanded for 8 days before downstream analysis.

## Epigenome editing in CD34+ cells
The mRNA transcription template was generated via PCR with primers (Supplementary Table 2) that introduce the dT7 promoter sequence and install a poly(A) tail, as previously described[62] and the products were purified by QIAGEN quick PCR purification kit. The mRNA was transcribed using the T7 High-Yield RNA kit (NEB, #E2040S) according to manufacturer instructions, with full substitution of N1-methylpseudouridine (Trilink) for uridine and co-transcriptional capping with CleanCap® Reagent AG (TriLink, #N-7113). The resulting mRNA was purified using the NEB Monarch® RNA Cleanup Kit (500 μg) (NEB, #T2050). CD34+ cells were mixed with 2 μg of epigenome editor mRNA and *HBG* promoter sgRNAs (150 pmol), electroporated with the

Lonza 4D-Nucleofector System (electroporation program DS-130), and incubated for 24 h in CD34+ cell maintenance medium. GFP+ or BFP+ cells were purified by FACS and transferred to phase 1 CD34+ cell differentiation medium.

## Bisulfite amplicon sequencing
HUDEP2 cells were harvested and genomic DNA extracted using the PureLink genomic DNA mini kit (Invitrogen, # K182002) according to manufacturer's instructions. Bisulfite conversion was performed using the EpiTect Fast Bisulfite kit (Qiagen, #59824) according to manufacturer instructions, and target sequences were amplified using Q5U (NEB, #M0515L). Primers were designed to amplify regions of the *HBG1* and *HBG2* 5′ UTR and promoters from bisulfite converted DNA with alternate primers designed where necessary to improve compatibility across the two genes. Primers used for the amplification of bisulfite-converted DNA are listed in Supplementary Table 2. Amplicons from each sample were pooled at an equimolar ratio before library preparation using the NEB Ultra II DNA library prep kit for Illumina and NEBNext multiplex oligos for Illumina (NEB) with 0.6× SPRIselect beads used for size selection and PCR cleanup. Libraries were sequenced with the Ramaciotti Center for Genomics using the MiSeq v2 2 × 250 flow cell (Illumina) and data were analyzed using the nf-core methylseq workflow (v2.5.0)[63] with deduplication skipped.

For analysis of TETv4 activity in relation to guide position for cells treated with individual guides, the percentage of methylated CpGs at each site in each sample was normalized to the mean methylation of mock-treated cells. Positions of CpG sites were then determined in reference to the center of the guide sequence used for the respective TETv4 treatment.

## Quantitative PCR
HUDEP2 cells were harvested and washed in 1× PBS before lysis and RNA isolation using TRI™ reagent (Sigma-Aldrich, #T9424) according to the manufacturer's instructions. RNA was purified using the RNeasy® Mini Kit and RNase-free DNase set (Qiagen, #74106) according to the manufacturer's instructions. cDNA was synthesized SuperScript™ VILO™ master mix (Invitrogen, #11755250) with a two-hour incubation period at 37 °C. cDNA samples were diluted 1:400 to optimize the dynamic range of qPCR for globin gene transcripts. qPCR was performed using PowerUp™ SYBR green reaction mix (Applied Biosystems, #A25778) in 384 well format on the ViiA7 (QuantStudio Real Time PCR software v1.3) or Quantstudio 7 Pro RT-PCR machine (QuantStudio 6/7 Pro software v1.8.0) (Applied Biosystems) with default settings. Primers used in qPCR analysis are listed in Supplementary Table 3. Expression of *HBG* was normalized to the sum of *HBG* and *HBB* expression (*HBG/(HBG + HBB)*).

## TETv4 Individual guide correlation analysis
Pearson's correlation coefficients were calculated in R between the relative expression of *HBG* (*HBG/(HBG + HBB)*) measured by qPCR, and methylation at CpG sites or the mean of sets of CpG sites measured by bisulfite sequencing.

## Fetal hemoglobin+ cell quantification by flow cytometry
HUDEP2 cells were harvested from culture or thawed from cryopreserved samples. Harvested cells were fixed in 0.05% glutaraldehyde (Sigma Aldrich, #G5882) and permeabilized with 0.1% Triton X-100 (Sigma Aldrich, #X100) before staining with HbF-APC antibody (Invitrogen, #MHFH05, clone HBF-1) diluted 1:20 in PBS with 0.1% bovine serum albumin (BSA) (Sigma Aldrich, #A4503) for 17 min. Stained cells were resuspended in PBS with 0.1% BSA and analyzed by flow cytometry using the LSRFortessa X-20 (BD Biosciences) with BD FACSDiva (v9.0) and FlowJo (v10.10) software. A representative gating strategy is shown in Supplementary Fig. 5.

## RNA sequencing

RNA for sequencing was extracted as described (see Quantitative PCR). Stranded mRNA prep (Illumina) sequencing libraries and RNA sequencing were performed by the Ramaciotti Centre for Genomics using NextSeq 1×75 bp high output or NovaSeq 2x100bp flow cells (Illumina). Sequencing data was processed using the nf-core RNAseq workflow (v3.14)[63] with the Kallisto pseudoaligner. Kallisto parameters were set with 100 bootstrap samples, and fragment length set to 75 with a standard deviation of 2, where single-end reads were used.

## Differential expression analysis

Differential expression analysis was performed using the Sleuth package (v0.30.1)[64] within the R software (v4.4.1). Briefly, transcript counts generated from Kallisto were read into R using sleuth_prep, aggregating on gene level (gene_mode FALSE), with the sample table designed for pairwise comparison to the control. Counts were transformed to use log base 2 ($log2(counts + 0.5)$) and were filtered to have at least 20 reads in one condition (e.g., for RNA-seq on the TETv4 experiments, at least 20 reads in 3 samples). After fitting the full model, which considered the TETv4 treatment used, and the reduced model (ignoring the treatment), the likelihood ratio test was performed using sleuth_lrt with the Benjamini-Hochberg false discovery rate used to correct for multiple comparisons. Estimated log2 fold changes were calculated by summing normalized counts (filtered) from every transcript to their respective gene for each sample, followed by log2 transformation as above. The mean counts for each condition were calculated, and the log2 fold change was calculated by subtracting the mean log2 count of the baseline condition from the test condition.

## Gene set enrichment analysis

Gene set enrichment analysis (GSEA) was performed using data from differential expression analysis in R. MSigDB pathways were obtained using the msigdbr package v7.5.1[65] in R. The filtered gene list was ranked by the estimated fold change and used as input for the GSEA function from clusterProfiler v4.12.6[66,67] with the eps parameter set to 0.

## Off-target analysis of *HBG* guides

Predicted off-target sites for each guide were obtained using the CRISPR design tool (Benchling) following the method of Hsu et al.[68]. The off-target regions were converted to a GRanges object in R and compared to a gene list in GRanges format using the precede and follow functions from the GenomicRanges package v1.58.0[69] in R to identify the nearest adjacent gene on either side of the target region. Identified genes for each guide were used to filter differential expression results from the respective TETv4 treatment, and genes discovered at an FDR of 5% in any of the six TETv4 guide treatments were analyzed across all TETv4 guide treatments to determine whether upregulation was guide-specific.

## CRISPR/Cas9 genome editing in MBD2

The PX458 plasmid encoding *Sp*Cas9, an sgRNA scaffold, and the *EGFP* gene was kindly donated by Feng Zhang[70] (Addgene plasmid, #48138).

For creating the MBD2Y178F mutation, a guide sequence was designed to cut between nucleotides +647 and +648 of *MBD2*. Oligonucleotides used for gene editing are listed in Supplementary Table 4. Single-stranded oligos containing the guide sequence were synthesized (IDT) with complementary sticky ends to the BbsI sites of PX458. The annealed guide fragment was cloned into PX458 digested with BbsI, and the cloning product was sequenced to confirm the final sgRNA sequence. A donor ssDNA oligo of 150 nucleotides containing the +643A>T mutation and complementary flanking sequences from *MBD2* was used to support HDR (Supplementary Table 4).

HUDEP2 cells were washed in 1× PBS before transfection using the Neon electroporation system (Invitrogen) with 1.6 µg PX458 and 10 pmol donor oligo per $5 \times 10^5$ cells in 1× buffer T. Each transfection was performed three times using single pulses of 1100, 1200, and 1300 V for 20, 30, and 40 ms, respectively. Wild-type controls were subjected to the same transfection conditions without a guide sequence in the PX458 plasmid.

Following 72 h of recovery from transfection, FACS was performed to pool live cells that had taken up PX458 (KO+/EGFP+) using the FACSMelody (BD Biosciences). After a further 96 h, pools were sorted to select live cells that no longer expressed PX458 (KO+/EGFP-), which were seeded as clonal populations for biological replication. Clones were screened by PCR and Sanger sequencing to confirm the presence or absence of the MBD2+643A>T mutation, and western blot to confirm equivalence of MBD2 expression to WT clones. An equal number of WT clones were selected for RNAseq analysis based on proximity to the mean expression of *HBG* ($HBG/(HBG+HBB)$) for this group in pilot experiments.

## Western blot

Harvested cells were washed with 1× PBS and incubated in 10× volume of buffer A (10 mM HEPES (Gibco, pH 7.9), 1.5 mM MgCl$_2$, 10 mM KCl) on ice for 10 min to lyse cells. The pelleted lysate was resuspended in 3× volume of buffer C (20 mM HEPES (pH 7.9), 420 mM NaCl, 1.5 mM MgCl$_2$, 0.2 mM EDTA, 25% glycerol) and incubated on ice for 20 min for nuclear lysis. Debris was pelleted by centrifugation, and the supernatant (nuclear extract) was recovered.

Nuclear extracts were made up in 1× Laemmli buffer (63 mM Tris-HCl, 100 mM DTT, 20 mg/ml SDS, 100 µg/ml bromophenol blue, 10% glycerol) and heated at 95 °C for 5 min for denaturation. Extracts were separated on a 10% NuPage® Bis-Tris gel (Invitrogen, #NP0301) in MOPS buffer (Invitrogen, #NP0001) and transferred onto nitrocellulose blotting membrane by electrophoresis in transfer buffer (25 mM Tris, 200 mM glycine, 20% v/v methanol). Blocking was performed overnight at 4 °C using 3.5% skim milk in tris-buffered saline with tween-20 (TBST, 50 mM Tris (pH 7.4), 150 mM NaCl, 0.1% v/v Tween 20 (Sigma-Aldrich, #P9416)) and washed with TBST before incubating for 1 h at room temperature with anti-MBD2 antibody (Abcam, #ab188474, clone EPR18361, 1:10,000 in TBST), UHRF1 antibody (Millipore, #MABE308,1:5000), BCL11A antibody (Abcam, #ab191401, 1:1000), Hemoglobin γ (Cell Signalling Technology, #39386 s, 1:3000). The membrane was washed in TBST before incubation with the secondary antibody (Abcam, #ab205718, 1:20,000 in TBST) for 1 h at room temperature. The membrane was washed again with TBST before imaging using the ImageQuant™ LAS 500 (Software v1.1.0, GE Healthcare) with Immobilon® western chemiluminescent HRP substrate (Merck). The membrane was stripped in stripping buffer (200 mM glycine, 0.1% SDS, 1% Tween 20, pH 2.2) before re-blocking and re-blotting with anti-β-actin antibody (Sigma-Aldrich, #A1978, clone AC-15, 1:20,000 in TBST) and secondary antibody (Cytiva, #NA931V, 1: 20,000 in TBST) as a loading control.

## CUT&RUN assay

The CUT&RUN assay was performed as described[71]. For each experimental condition, ~$5 \times 10^5$ cells were collected, incubated with primary antibody (Supplementary Table 5) at 1:100 dilution in antibody buffer at 4 °C for 2 h, then labeled with protein A conjugated micrococcal nuclease for 1 hour. Digestion was initiated by adding CaCl$_2$ to a final concentration of 2 mM and incubating at 0 °C on a cold block for 1 hour, followed by DNA purification with Phenol-chloroform. Library construction for Next Generation DNA sequencing (NGS) was performed using an NEBNext Ultra II DNA Library Prep kit from NEB (E7645S). Indexed samples were sequenced by paired-end NGS, using an Illumina NextSeq Mid-Output (150 cycles) kit or Novaseq (100 cycles) kit. FASTQ files were mapped to hg19 using BWA-MEM (v.0.7.16a). Reads that could not be uniquely mapped to the human genome were removed by SAMtools (v.0.17). Peaks were called by

using MACS2 (v.2.1.1). BigWiggle files were generated using DeepTools (v.3.2.0). The codes used to perform CUT&RUN (HemTools cut_run) are available at https://github.com/YichaoOU/HemTools and https://doi.org/10.5281/zenodo.4783657. Pipeline documentation is available at https://hemtools.readthedocs.io/en/latest/.

## Electrophoretic Mobility Shift Assay (EMSA)

The MBD2sc-pET32a plasmid containing the MBD, IDR, and CC of MBD2 fused to the CC domain of GATAD2A was a kind gift from David Williams[72]. A modified variant containing the Y178F mutation was created by overlap extension PCR and RE cloning to introduce an A>T substitution at position +587 relative to the start codon.

WT and Y178F constructs were transformed into the Rosetta *E. coli* strain and expression was induced with 0.25 mM IPTG overnight at 18 °C. Cells were resuspended in lysis buffer (10 mM Tris-HCl pH 7.5, 300 mM NaCl, 10 mM Imidazole) supplemented with EDTA-free protease inhibitor (Roche, #11836170001) and sonicated using a Branson 450 sonifier with a 3 mm probe at 40% amplitude for 10 cycles of 45 s on/off. Lysates were clarified by centrifugation at $12,000 \times g$ for 30 min at 4 °C before purification using HisPur™ Ni-NTA Spin Columns (Thermo Fisher Scientific, #88225) according to the manufacturer's instructions. Elution fractions were concentrated and exchanged into storage buffer (10 mM Tris-HCl pH 7.5, 150 mM NaCl, 1 mM EDTA, 1 mM PMSF) using Amicon Ultra 30 kDa centrifugal filters (Millipore #UFC9030) according to the manufacturer's instructions. Protein concentrations were measured by nanodrop spectroscopy using the Gill and von Hippel method to estimate $\varepsilon$ before adding 10% glycerol for storage. Purified proteins were run on 4–12% NuPage® Bis-Tris gel (Invitrogen, #NP0322) as described (see "Western blot") and stained with QC colloidal Coomassie stain (Bio-Rad) according to the manufacturer's instructions to confirm protein samples were equivalent before use in EMSA.

Oligonucleotides used in radiolabelled probes are listed in Supplementary Table 6. The antisense strand for each probe was labeled with P-32 from γ-32P ATP (Revvity, #BLU502A100UC) using T4 PNK (NEB, #M0201S), before annealing the sense strand by slow cooling from 100 °C to room temperature. The annealed probes were purified using quick spin columns for radiolabelled DNA purification (Roche, #G25DNA-RO). Antibody for MBD2 (Abcam, #ab188474, clone EPR18361) was used as indicated to confirm the presence of MBD2 in protein-probe complexes. Purified MBD2sc or MBD2sc Y178F protein was mixed with probe at a 50× molar excess and allowed to complex for 10 min at room temperature. Complexed samples were loaded on 6% native polyacrylamide gel in TBE buffer (45 mM Tris, 45 mM boric acid, 1 mM EDTA). Electrophoresis was performed at 4 °C and 250 V for 1 h and 40 min, and vacuum dried before exposing a FUJIFILM BAS Cassette2 phosphor screen overnight. Imaging was performed using the GE Typhoon FLA 9500 fluorescent image analyzer.

## Statistics and reproducibility

No statistical methods were used to predetermine the sample size. Sample sizes were selected to ensure that all experimental and control samples prepared in parallel could be processed to a high standard with appropriate replication. Experiments using the UHRF1-AID system, including Western blot, were replicated two times using independently derived UHRF1-AID HUDEP2 clones. Representative data from one clone is presented. Reproduction of EMSA results was not attempted. The success of the EMSA experiment was determined by the replication of a positive control probe from the literature, which has been shown to bind the protein of interest.

No data were excluded from the analysis. Where clonal populations of modified cells were produced, screening was performed as described to select appropriate clones prior to analysis. The experiments were not randomized. The investigators were not blinded to allocation during experiments and outcome assessment.

## Reporting summary

Further information on research design is available in the Nature Portfolio Reporting Summary linked to this article.

## Code availability

The codes used to analyze ATAC-seq data (HemTools atac_seq) and CUT&RUN data. (HemTools cut_run) are available at https://github.com/YichaoOU/HemTools and https://doi.org/10.5281/zenodo.4783657. The Pipeline documentation is available at https://hemtools.readthedocs.io/en/latest/.

## Data availability

Raw next-generation sequencing data generated in this project have been submitted to GEO. The RNA-seq data generated in this study have been deposited in the GEO database under accession code GSE284289, https://www.ncbi.nlm.nih.gov/geo/query/acc.cgi?acc=GSE284289.
The ATAC-seq data generated in this study have been deposited in the GEO database under accession code GSE284290, https://www.ncbi.nlm.nih.gov/geo/query/acc.cgi?acc=GSE284290. The CUT&RUN data generated in this study have been deposited in the GEO database under accession code GSE284291, https://www.ncbi.nlm.nih.gov/geo/query/acc.cgi?acc=GSE284291. Source data are provided with this paper.

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

## Acknowledgements

PX458 plasmid was a gift from Feng Zhang, Broad Institute (Addgene #48138). pcDNA3-mCherry plasmid was a gift from Beeke Weinert. MBD2sc-pET32a was a gift from David Williams Jr., the University of North Carolina at Chapel Hill. HUDEP2 cells were kindly provided by Ryo Kurita and Yukio Nakamura, RIKEN BioResource Center, Japan. The authors acknowledge Emma Johansson Beves and Mayssa Sharabas from the Biological Resources Imaging Laboratory (Mark Wainwright Analytical Centre, UNSW Sydney) for assistance with flow cytometry. This research includes computations using the computational cluster Katana, supported by Research Technology Services at UNSW Sydney. The authors also thank Xuili An (New York Blood Center) for APC-conjugated anti-BAND3 and staff at St. Jude Institutional Core Facilities, including the Center for Advanced Genome Engineering, the Flow Cytometry and Cell Sorting Shared Resource and the Hartwell Center. The authors also thank David Cullins and Emilia Kooienga from the Experimental Hematology Flow Cytometry and Cell Sorting core. This work was supported by an Australian National Health and Medical Research Council grant 2020861 (to M.C. and K.G.R.Q.), National Institutes of Health (NIH) grants U01 HL163983 (to M.J.W.), R01 156647 (to M.J.W.), K01DK132453 (to P.A.D.), the Gates Foundation (to M.J.W.), and A.L.S.A.C. H.W.B. and M.F.O. were supported by Australian Government Research Training Program (RTP) postgraduate scholarships. T.M. was supported by the Government of India DBT Ramalingaswami Re-entry Fellowship BT/RLF/Re-entry/42/2022.

## Author contributions

H.W.B. and R.F. contributed equally. M.J.W. and M.C. contributed equally. Y.C., K.G.R.Q., M.J.W., and M.C. jointly supervised research. H.W.B., R.F., J.P.M., K.G.R.Q., M.J.W., and M.C. conceived and designed the experiments. H.W.B., R.F., Y.Y., J.D., P.A.D., T.M., and J.Z. performed the experiments. H.W.B., R.F., and M.S. performed statistical analysis. H.W.B., R.F., M.S., M.F.O., Y.L., Y.W., and Y.C. analyzed the data. P.A.D. and Y.L. contributed analysis tools. H.W.B., R.F., K.G.R.Q., M.J.W., and M.C. wrote the manuscript with input from all authors.

## Competing interests

M.J.W. owns equity in Cellarity Inc. and is a consultant for Fulcrum Pharmaceuticals. The authors declare that they have no other competing interests.
