## [Peer Review file · Nature Communications]

Removal of promoter CpG methylation by epigenome editing reverses *HBG* silencing

Corresponding Author: Professor Merlin Crossley

Version 0:

Reviewer comments:

Reviewer #1

(Remarks to the Author)

The manuscript by Henry W. Bell et al. identified the DNMT1-associated maintenance methylation protein UHRF1 as a mediator of HBG repression in a genome-wide screen. Loss of UHRF1 in the adult-type erythroid cell line HUDEP2 caused global demethylation and HBG activation that was reversed upon localized promoter remethylation. Conversely, targeted demethylation of the HBG promoters activated their genes in HUDEP2 or primary CD34+ cell-derived erythroblasts. Mutation of MBD2, a CpG methylation reading component of the NuRD complex, recapitulated the effects of promoter demethylation. This study demonstrates that localized CpG-methylation at the HBG promoters facilitates gene silencing, and identify a potential therapeutic approach for b-hemoglobinopathies via epigenomic editing. Despite the interesting data and novel findings presented by the authors, some comments and questions have been raised to further solidify their conclusions.

1. Hudep-2 basically does not express gamma-globin, so why is the positive rate of F cells in aavs1 36.5% in Figure 1f?
2. In Figure 4d, why did the expression of HBA and HBD also change? Could the author explain the reason?
3. What is the efficiency of targeted methylation? The article does not mention it. as far as I know, the electroporation efficiency of HUDEP-2 is not high. How did the author manage to electroporate the targeted demethylation vector into HUDEP-2 and achieve such a high efficiency?
4. The article mentions that "in contrast, the level of H3K4me3 decreased at the HBB promoter. Interestingly, localization of the HBG repressor protein BCL11A, which functions at least partly through interactions with the NuRD co-repressor complex, was not reduced at the HBG promoters upon TETv4 treatment". However, this is not shown in the figure.
5. Mutating the MBD2 site only indicates the function of the MBD2 domain, and does not prove that CpG methylation may be required for the activity or recruitment of the NuRD complex. More experiments should be performed to demonstrate this, such as methylation EMSA.

Reviewer #2

(Remarks to the Author)

Bell et al. investigate the role of CpG methylation in the regulation of gamma-globin (HBG) expression. Induction of hemoglobin F (HbF, where HBG are the beta-like globin chains), is the ultimate goal in therapy of beta hemoglobinopathies. Gene therapy approaches to attain this have recently been approved by the FDA but for the majority of patients worldwide this therapeutic avenue is inaccessible. Epigenetic therapies may be more accessible and potentially safer than 'conventional' gene therapy. With this in mind, the authors of this manuscript re-visit the relevance of HBG promoter methylation in the regulation of HBG gene expression. It's been known for many decades that demethylating drugs such as 5-azacytidine lead to induction of HBG. However, the precise mechanism of action, and whether direct or indirect, remained controversial, a result of the broad effects of these drugs. Here the authors make use of recent technical advances that use dCas9 to target specific CpGs for either methylation or demethylation. Using multiple complementary experimental approaches, including rigorous controls, they show that 6 CpGs in the promoter of HbG are each individually and additively responsible for repressing HBG expression when methylated, and that their demethylation is sufficient to induce HBG. Induction is independent of the principal repressor of this gene, BCL11A, and is long-lasting. This work therefore opens a new front in potential therapies for beta hemoglobinopathies. It also is of interest in analyzing the effect of methylation status of a handful, specific CpGs on gene expression, an area that is not yet settled science.

The authors first focused on HBG promoter methylation when a CRISPR/Cas9 screen identified UHRF1, a binding partner of DNMT1, as a negative regulator of HBG. The authors used dCas9-DNMT3a and dCas9-Tet to targeted specific CpGs in the promoter for either methylation or demethylation respectively, and showed the direct and additive negative effect of their methylation on HBG expression. The authors used both HUDEP2 cell line and primary erythroblasts derived from CD34+ bone marrow progenitors for these experiments. Chromatin analysis showed that demethylation status of these CpGs was sufficient to bring about changes in chromatin consistent with transition from a repressive chromatin environment to an active one.

The only somewhat weak part of this otherwise elegant paper is the implication that MBD2 mediates the repressive effects of the methylated CpGs at the HBG promoter. The authors don't find any transcription factor binding sites directly at the CpG sites, and so they hypothesize that MBD2 binds methylated CpGs and recruits the repressive NURD complex. They introduce (and presumably overexpress) an MBD2 mutant that fails to bind CpGs and show that its presumed negative competitive activity with native MBD2 leads to induction of HBG. Unlike their other experiments, however, this experiment is less easy to interpret since it both relies on overexpression and does not implicate the HBG promoter CpGs specifically. I would recommend either discussing the drawbacks of this experiment explicitly in the paper, omitting the data from the paper, or expanding the experiment to include epistatic analysis of CpG methylation effects (or lack thereof) in the presence of the MBD2 mutation.

Reviewer #3

(Remarks to the Author)

Reviewer #4

(Remarks to the Author)

In this very interesting manuscript, Bell et al present convincing data from direct lines of evidence that methylation of CpGs in the human HBG proximal promoter make a major contribution to silencing of the gene in adult erythroid cells. In addition to identifying UHRF-1 as a silencer of HBG via its critical key role in DNMT1-mediated maintenance methylation, the authors show that removal of methylation at specific sites in the HBG promoter activate transcription at a high level and that re-methylation of these specific sites restores silencing. This work provides definitive evidence for the long debated role of site specific promoter methylation in embryonic/fetal globin gene silencing and supports previous evidence for the role of the methyl cytosine binding protein MBD2 in enforcing the methylation induced silencing. This work also raises the possibility of editing HBG promoter methylation as a site specific therapeutic avenue for treatment of beta thalassemia and sickle cell anemia.

Overall, the experiments are of high quality and support the interpretation and conclusions of the authors. By utilizing multiple lines of experimental evidence the conclusions are strengthened.

There are some issues and questions that if addressed would clarify the data interpretation and greatly strengthen the conclusions in the manuscript

1. In Fig 2, since there is not complete demethylation of promoter CpGs in the UHRF-1 KO HUDEP-2 cells, it would be informative to see the allele distribution of the demethylated sites in addition to overall percent demethylation data. ie. in the setting of ~50% methylation of most sites overall are there some alleles with most or all sites demethylated?
2. In Fig 2B the ATAC Seq and GATA1 peaks are fairly hard to discern in the CUT & RUN data. This could be improved by including an inset with an enlarged view of those areas in the plots.
3. In Fig 3 the colors of the line graphs for the TETv4HBG. and dTETv4HBG are hard to distinguish. Using more contrasting colors or using different symbols for the data points on those graphs would improve the ability to see the differences.
4. In Fig 5, panel e there is no plot for the CUT & RUN data for BCL11A although it is discussed in the results section. Although such data for BCL11A is shown in Fig 2b. it is important to show it for the completely demethylated loci as well. In Fig 5e it appears that the CUT & RUN signals for ATAC Seq GATA-1, NFY, and H3K4me3 are higher in the dead TETv4HBG samples, which is contrary to the overall data of the manuscript. I suspect this was an inadvertent transposition of the graphics, given the overall strength of the data elsewhere supporting the increase in these signals with promoter demethylation concomitant with increased HBG expression.
5. Regarding CUT & RUN data, this reviewer could not find the description of the methodology and antibody reagents used for these assays.
6. The model in Figure 7 is overall quite instructive and clear, but based on the data in Fig 2e and the results discussion

section regarding the CUT & RUN results in Fig 5e, it seems that BCL11A is bound to the proximal promoter when the CpG sites are demethylated and the gene is actively transcribed rather than it being unbound as shown in the model depiction.

Version 1:

Reviewer comments:

Reviewer #1

(Remarks to the Author)

The authors have addressed most of my questions; however, I still have one concern. In Figures 2c and 5e, the enrichment of BCL11A at the HBG locus appears to be higher compared to the control group. Given that BCL11A is a potent repressor of HBG expression, how can the levels of HBG increase so significantly following UHRF1 knockout or TETv4 treatment, despite the increased BCL11A binding? How do the authors explain this apparent discrepancy?

Reviewer #2

(Remarks to the Author)

The modifications and additions to Figure 6 are excellent. The authors have responded to all my concerns.

Reviewer #3

(Remarks to the Author)

Reviewer #4

(Remarks to the Author)

In this revised manuscript, Bell et al provide convincing evidence that CpG methylation in the HBG promoters is critical in maintaining transcriptional silencing of these genes during adult stage erythropoiesis. By using recently developed gene editing techniques this work directly shows that removal of the specific methylation sites in the HBG promoters results in transcriptional activation despite continued occupancy of the major developmental repressor of these genes, BCL11A. Moreover they demonstrate that addition of methylation at the HBG promoters results in silencing. In addition their results are in concert with recent data showing that the methylcytosine binding preference of MBD2 is required to read the methylation marks and execute silencing of the HBG genes through the MBD2-NuRD chromatin remodeling complex.

The authors have fully and adequately addressed all of the issues raised in this reviewer's critique of the initial manuscript.

Version 2:

Reviewer comments:

Reviewer #1

(Remarks to the Author)

The authors have addressed my concerns.

Reviewer #1 (Remarks to the Author):

The manuscript by Henry W. Bell et al. identified the DNMT1-associated maintenance methylation protein UHRF1 as a mediator of HBG repression in a genome-wide screen. Loss of UHRF1 in the adult-type erythroid cell line HUDEP2 caused global demethylation and HBG activation that was reversed upon localized promoter remethylation. Conversely, targeted demethylation of the HBG promoters activated their genes in HUDEP2 or primary CD34⁺ cell-derived erythroblasts. Mutation of MBD2, a CpG methylation reading component of the NuRD complex, recapitulated the effects of promoter demethylation. This study demonstrates that localized CpG-methylation at the HBG promoters facilitates gene silencing, and identify a potential therapeutic approach for b-hemoglobinopathies via epigenomic editing. Despite the interesting data and novel findings presented by the authors, some comments and questions have been raised to further solidify their conclusions.

1. Hudep-2 basically does not express gamma-globin, so why is the positive rate of F cells in aavs1 36.5% in Figure 1f?

Response: This is a simple misunderstanding and we have clarified the text to make it clearer. Figure 1 panels d-f actually show results in CD34⁺ cell-derived erythroblasts, which do have background of HbF expression. Unfortunately only the figure legend had this information. We have now modified the text to include this information also:

Top of page 4: “Disruption of *UHRF1* in CD34⁺ HSPCs, followed by in vitro differentiation, raised *HBG* and HbF in erythroid progeny to nearly the same levels as disruption of the *BCL11A* erythroid enhancer according to a clinically approved strategy¹⁴ (Fig. 1d-f and Extended Data Fig. 1a)”

2. In Figure 4d, why did the expression of HBA and HBD also change? Could the author explain the reason?

Response: This is an interesting point. The reason for the modest effects on the expression of non-target genes including HBD and HBA as a result of TETv4 treatment targeting HBG is not entirely clear but may be related to a propensity for a low level of spontaneous erythroid differentiation in the HUDEP2 model after transfection.

On page 7 and 8: We have reworked the results paragraph exploring non-target gene expression in an effort to cover this issue.

3. What is the efficiency of targeted methylation? The article does not mention it. as far as I know, the electroporation efficiency of HUDEP-2 is not high. How did the author manage to electroporate the targeted demethylation vector into HUDEP-2 and achieve such a high efficiency?

Response: The reviewer is correct that overall rates of transfection are not high but high rates of epigenetic modification in HUDEP2 and CD34+ cells were achieved by first FACS sorting of the transfected cells for fluorescent marks expressed with the editor. We have added references to the purification of successfully transfected cells to the relevant results sections on page 5, as well as adding additional emphasis to the purification step in the epigenome editing methods sections to help the reader more readily appreciate this aspect of the methodology.

4. The article mentions that “in contrast, the level of H3K4me3 decreased at the HBB promoter. Interestingly, localization of the HBG repressor protein BCL11A, which functions at least partly through interactions with the NuRD co-repressor complex, was not reduced at the HBG promoters upon TETv4 treatment”. However, this is not shown in the figure.

Response: This is a good point. We have now added BCL11A CUT&RUN data to modified Figure 5e.

5. Mutating the MBD2 site only indicates the function of the MBD2 domain, and does not prove that CpG methylation may be required for the activity or recruitment of the NuRD complex. More experiments should be performed to demonstrate this, such as methylation EMSA.

Response: This is a good suggestion. We have now completed and included an EMSA in Figure 6 and adjusted the discussion of the result as below.

On page 9: We have improved our rationale for targeting MBD2 in this way, first by emphasizing that the existing literature links the importance of MBD2 in *HBG* regulation to its function in the NuRD complex, and second, by extending our explanation that precisely targeting this residue in MBD2 allows us to isolate and study whether MBD2's ability to bind CpG methylation is important for *HBG* regulation. In addition, we have added an EMSA in Figure 6c validating that the MBD2 Y178F mutation inhibits binding to CpG sites including those found in the *HBG* promoter.

Reviewer #2 (Remarks to the Author):

Bell et al. investigate the role of CpG methylation in the regulation of gamma-globin (HBG) expression. Induction of hemoglobin F (HbF, where HBG are the beta-like globin chains), is the ultimate goal in therapy of beta hemoglobinopathies. Gene therapy approaches to attain this have recently been approved by the FDA but for the majority of patients worldwide this therapeutic avenue is inaccessible. Epigenetic therapies may be more accessible and potentially safer than 'conventional' gene therapy. With this in mind, the authors of this manuscript re-visit the relevance of HBG promoter methylation in the regulation of HBG gene expression. It's been known for many decades that demethylating drugs such as 5-azacytidine lead to induction of HBG. However, the

precise mechanism of action, and whether direct or indirect, remained controversial, a result of the broad effects of these drugs. Here the authors make use of recent technical advances that use dCas9 to target specific CpGs for either methylation or demethylation. Using multiple complementary experimental approaches, including rigorous controls, they show that 6 CpGs in the promoter of HbG are each individually and additively responsible for repressing HbG expression when methylated, and that their demethylation is sufficient to induce HbG. Induction is independent of the principal repressor of this gene, BCL11A, and is long-lasting. This work therefore opens a new front in potential therapies for beta hemoglobinopathies. It also is of interest in analyzing the effect of methylation status of a handful, specific CpGs on gene expression, an area that is not yet settled science.

The authors first focused on HbG promoter methylation when a CRISPR/Cas9 screen identified UHRF1, a binding partner of DNMT1, as a negative regulator of HbG. The authors used dCas9-DNMT3a and dCas9-Tet to targeted specific CpGs in the promoter for either methylation or demethylation respectively, and showed the direct and additive negative effect of their methylation on HbG expression. The authors used both HUDEP2 cell line and primary erythroblasts derived from CD34+ bone marrow progenitors for these experiments. Chromatin analysis showed that demethylation status of these CpGs was sufficient to bring about changes in chromatin consistent with transition from a repressive chromatin environment to an active one.

The only somewhat weak part of this otherwise elegant paper is the implication that MBD2 mediates the repressive effects of the methylated CpGs at the HbG promoter. The authors don't find any transcription factor binding sites directly at the CpG sites, and so they hypothesize that MBD2 binds methylated CpGs and recruits the repressive NURD complex. They introduce (and presumably overexpress) an MBD2 mutant that fails to bind CpGs and show that its presumed negative competitive activity with native MBD2 leads to induction of HbG. Unlike their other experiments, however, this experiment is less easy to interpret since it both relies on overexpression and does not implicate the HbG promoter CpGs specifically. I would recommend either discussing the drawbacks of this experiment explicitly in the paper, omitting the data from the paper, or expanding the experiment to include epistatic analysis of CpG methylation effects (or lack thereof) in the presence of the MBD2 mutation.

Response: We have now worked to clarify our rationale in the MBD2 experiments. Most importantly, we have now emphasised that these experiments were performed by modifying endogenous MBD2 using CRISPR/Cas9 genome editing rather than overexpression, though a western blot was still necessary to confirm that genome

editing had not affected protein expression in any way that was not detected by sequencing.

On page 9-10: The section on introducing MBD2 Y178F to HUDEP2 cells has been modified to clarify the methodology, highlighting that endogenous MBD2 was mutated in HUDEP2 cells rather than MBD2 overexpression. In addition, we have expanded the analysis of other genes discovered in our differential expression analysis, highlighting those with known or suspected roles in *HBG* regulation in the text and on Figure 6d with discussion of how their differential expression would be expected to influence *HBG* in our results. Finally, we have added an EMSA to Figure 6c demonstrating that the MBD2 Y178F mutation inhibits binding to methylated CpG sites such as those in the *HBG* promoter, in support of the direct mechanism we propose. Together these changes improve both the interpretability and the strength of our conclusions.

Reviewer #3 (Remarks to the Author):

Reviewer #4 (Remarks to the Author):

In this very interesting manuscript, Bell et al present convincing data from direct lines of evidence that methylation of CpGs in the human *HBG* proximal promoter make a major contribution to silencing of the gene in adult erythroid cells. In addition to identifying UHRF-1 as a silencer of *HBG* via its critical key role in DNMT1-mediated maintenance methylation, the authors show that removal of methylation at specific sites in the *HBG* promoter activate transcription at a high level and that re-methylation of these specific sites restores silencing. This work provides definitive evidence for the long debated role of site specific promoter methylation in embryonic/fetal globin gene silencing and supports previous evidence for the role of the methyl cytosine binding protein MBD2 in enforcing the methylation induced silencing. This work also raises the possibility of editing *HBG* promoter methylation as a site specific therapeutic avenue for treatment of beta thalassemia and sickle cell anemia.

Overall, the experiments are of high quality and support the interpretation and conclusions of the authors. By utilizing multiple lines of experimental evidence the conclusions are strengthened.

There are some issues and questions that if addressed would clarify the data interpretation and greatly strengthen the conclusions in the manuscript

1. In Fig 2, since there is not complete demethylation of promoter CpGs in the UHRF-1 KO HUDEP-2 cells, it would be informative to see the allele distribution of the demethylated sites in addition to overall percent demethylation data. ie. in the setting of ~50% methylation of most sites overall are there some alleles with most or all sites demethylated?

Response: This is a helpful point. These results are now shown in revised Figure 2b and discussed in the revised text:

Bottom of page 4: “Five days after disruption of *UHRF1* via transient expression of ribonucleoprotein (RNP) consisting of Cas9 and targeting sgRNA, overall methylation at the *HBG* promoter was reduced by approximately 50%, with 34% of alleles becoming demethylated at all six CpG sites (Fig. 2a, b).”

2. In Fig 2B the ATAC Seq and GATA1 peaks are fairly hard to discern in the CUT & RUN data. This could be improved by including an inset with an enlarged view of those areas in the plots.

Response: Yes, it is not clear why the GATA1 peak is modest. So that it can be seen we more clearly we have highlighted the *HBG1-HBG2* loci region and added an enlarged view in modified Figure2c.

3. In Fig 3 the colors of the line graphs for the TETv4HBG. and dTETv4HBG are hard to distinguish. Using more contrasting colors or using different symbols for the data points on those graphs would improve the ability to see the differences.

Response: Yes, we have made improvements in colour selection to increase contrast for all graphs that use 6 or more colours. In addition, since figure 3b is particularly difficult due to many overlapping lines, we have augmented this by reducing the number of colours, making dTETv4 lines a lighter shade of their TETv4 counterparts, and by implementing shapes for these points.

4. In Fig 5, panel e there is no plot for the CUT & RUN data for *BCL11A* although it is discussed in the results section. Although such data for *BCL11A* is shown in Fig 2b. it is important to show it for the completely demethylated loci as well. In Fig 5e it appears that the CUT & RUN signals for ATAC Seq GATA-1, NFY, and H3K4me3 are higher in the dead TETv4HBG samples, which is contrary to the overall data of the manuscript. I suspect this was an inadvertent transposition of the graphics, given the overall strength of the data elsewhere supporting the increase in these signals with promoter demethylation concomitant with increased HBG expression.

Response: Thank you. We added BCL11A CUT&RUN data to revised Figure 5e. Indeed, TETv4HBG and dTETv4HBG were mistakenly transposed in the original panel. We corrected this error in the revised Figure. Thank you for detecting this.

5. Regarding CUT & RUN data, this reviewer could not find the description of the methodology and antibody reagents used for these assays.

Response: We now added CUT&RUN methods to the Methods section (page 25) of the revised manuscript. New Supplementary Table 5 (Page 55) describes the antibodies used for CUT&RUN analysis.

6. The model in Figure 7 is overall quite instructive and clear, but based on the data in Fig 2e and the results discussion section regarding the CUT & RUN results in Fig 5e, it seems that BCL11A is bound to the proximal promoter when the CpG sites are demethylated and the gene is actively transcribed rather than it being unbound as shown in the model depiction.

Response: Thank you for noting this discrepancy in our attempt to present a simple model.

In Figure 7: we have revised the model to show that activators can bind to the demethylated promoter without necessarily preventing repressor binding. The improved figure also includes ZBTB7A which was discussed but not shown. On page 12, in the section discussing the model we have included reference to the fact that accessible chromatin as a result of reduced NuRD activity may also help binding of repressors as seen in the BCL11A CUT&RUN. Other changes in the discussion have been made to improve the precision of our discussion on how methylation works in concert with repressors to recruit MBD2-NuRD with reference to the literature.

REVIEWER COMMENTS

Reviewer #1 (Remarks to the Author):

The authors have addressed most of my questions; however, I still have one concern. In Figures 2c and 5e, the enrichment of BCL11A at the HBG locus appears to be higher compared to the control group. Given that BCL11A is a potent repressor of HBG expression, how can the levels of HBG increase so significantly following UHRF1 knockout or TETv4 treatment, despite the increased BCL11A binding? How do the authors explain this apparent discrepancy?

Thank you for highlighting this important point. We appreciate the apparent inconsistency that transcriptional activation of the HBG locus was associated with increased occupancy of the BCL11A repressor, as detected by CUT&RUN. Similar findings are reported by others (see PMID: 39607926 and PMID: 31245868). Comparatively weak detection of BCL11A at the repressed locus may be due to a closed chromatin state that reduces accessibility of BCL11A or the anti-BCL11A CUT&RUN antibody to bound DNA. We interpret the de-repression of *HBG* in the continued presence of BCL11A as evidence that, although BCL11A is known to be required for *HBG* repression, it is not sufficient to fully repress *HBG* in the absence of CpG methylation. We modified the text of the main manuscript to clarify this point as follows.

On page 4, we added the following line describing the BCL11A results in Figure 2c: “In agreement with previous reports, activation of HBG was associated with increased occupancy of the transcriptional repressor BCL11A, as detected by CUT&RUN (see discussion).”

Similarly, on page 9, we modified the text referring to Figure 5e:

“Interestingly, localisation of the *HBG* repressor protein BCL11A, which functions at least partly through interactions with the NuRD co-repressor complex^{18–20}, appeared to be increased at the *HBG* promoters upon TETv4 treatment, aligning with the effects of *UHRF1* disruption (see Figure 2c). Thus, TETv4 directed CpG demethylation of the *HBG* promoters induce an active transcriptional state without impairing the binding of the key repressor BCL11A.”

We added the following paragraph to the discussion (page 11):

Interestingly, activation of *HBG* by UHRF1 depletion or TETv4-directed promoter demethylation was associated with increased occupancy by the transcriptional

repressor BCL11A. The apparent inconsistency of increased repressor binding at an active gene may be explained by improved access of BCL11A (or its CUT&RUN antibody) to the open chromatin of the transcriptionally active promoter^{16,17}. The persistence of gene expression in this instance indicates that binding of BCL11A to the promoter is insufficient to fully repress transcription. Observations that BCL11A occupancy persists at the *HBG* promoters despite reactivation of the genes by CpG demethylation or ZBTB7A disruption¹⁶, and that BCL11A depletion combined with ZBTB7A depletion or CpG demethylation induces transcription additively^{9,36}, suggest that no single mechanism is dominant and that numerous approaches alone or in combination can be used to induce HbF therapeutically.

Reviewer #2 (Remarks to the Author):

The modifications and additions to Figure 6 are excellent. The authors have responded to all my concerns.

Thank you for your valuable input to improve our manuscript.

Reviewer #3 (Remarks to the Author):

Thank you for your involvement in reviewing our manuscript.

Reviewer #4 (Remarks to the Author):

In this revised manuscript, Bell et al provide convincing evidence that CpG methylation in the HBG promoters is critical in maintaining transcriptional silencing of these genes during adult stage erythropoiesis. By using recently developed gene editing techniques this work directly shows that removal of the specific methylation sites in the HBG promoters results in transcriptional activation despite continued occupancy of the major developmental repressor of these genes, BCL11A. Moreover they demonstrate that addition of methylation at the HBG promoters results in silencing. In addition their results are in concert with recent data showing that the methylcytosine binding preference of MBD2 is required to read the methylation marks and execute silencing of the HBG genes through the MBD2-NuRD chromatin remodeling complex.

The authors have fully and adequately addressed all of the issues raised in this reviewer's critique of the initial manuscript.

Thank you for your time and effort in reviewing our manuscript.